# Federated Dialogue-Semantic Diffusion for Emotion Recognition under Incomplete Modalities

**Xihang Qiu**[1,2,*], **Jiarong Cheng**[1,2,*], **Yuhao Fang**[1], **Wanpeng Zhang**[2],
**Yao Lu**[1], **Ye Zhang**[1,2], **Chun Li**[1,†]

[1] Shenzhen MSU-BIT University
[2] Beijing Institute of Technology
qiuxh@bit.edu.cn, lf_cyan27@outlook.com, lichun2020@smbu.edu.cn

## Abstract

Multimodal Emotion Recognition in Conversations (MERC) enhances emotional understanding through the fusion of multimodal signals. However, unpredictable modality absence in real-world scenarios significantly degrades the performance of existing methods. Conventional missing-modality recovery approaches, which depend on training with complete multimodal data, often suffer from semantic distortion under extreme data distributions, such as fixed-modality absence. To address this, we propose the Federated Dialogue-guided and Semantic-Consistent Diffusion (FedDISC) framework, pioneering the integration of federated learning into missing-modality recovery. By federated aggregation of modality-specific diffusion models trained on clients and broadcasting them to clients missing corresponding modalities, FedDISC overcomes single-client reliance on modality completeness. Additionally, the DISC-Diffusion module ensures consistency in context, speaker identity, and semantics between recovered and available modalities, using a Dialogue Graph Network to capture conversational dependencies and a Semantic Conditioning Network to enforce semantic alignment. We further introduce a novel Alternating Frozen Aggregation strategy, which cyclically freezes recovery and classifier modules to facilitate collaborative optimization. Extensive experiments on the IEMOCAP, CMUMOSI, and CMUMOSEI datasets demonstrate that FedDISC achieves superior emotion classification performance across diverse missing modality patterns, outperforming existing approaches.

## 1 Introduction

Multimodal Emotion Recognition in Conversations (MERC) [1, 2] has emerged as a pivotal technology of affective computing for understanding complex emotion states through synergistic fusion of textual, acoustic, and visual signals [3, 4, 5]. While State-of-the-art (SOTA) methods achieve remarkable performance under ideal multimodal conditions [6, 7], their efficacy collapses catastrophically in real-world scenarios plagued by unpredictable missing modalities caused by sensor failures, environmental noise, or privacy constraints [8, 9, 10].

Missing modality challenges in MERC can be categorized into two scenarios [11]: random missing protocol and fixed missing protocol. To recovery the missing information, researchers have proposed various recovery methods, primarily falling into two paradigms: **1) Latent space semantic recovery**: Lian et al. [12] propose GCNet to construct cross-modal correlations in the latent space through graph neural networks, leveraging temporal and speaker information from available modalities to

---

*Equal contribution.
†Corresponding authors.

39th Conference on Neural Information Processing Systems (NeurIPS 2025).

compensate for missing features. **2) Explicit modality recovery**: Wang et al. [8] leverage IMDer with a score-based diffusion model to directly reconstruct missing modality features, with available modalities guiding the reverse diffusion process as conditional inputs.

However, these methods struggle severely in fixed missing protocol scenarios, where a modality is entirely absent in local datasets. The lack of distributional priors for the missing modality and cross-modal alignment supervision leads to critical failures [13, 14]. Latent space recovery suffers from feature confusion due to missing modality-specific representations [15, 16], while generative modality recovery models fail catastrophically due to the absence of raw data required for the training of generative model. This highlights the inherent contradiction between current modality recovery techniques' reliance on modality completeness and their real-world applicability under extreme incomplete modalities.

This paper proposes the **Fed**erated **Di**alogue-Guided and **S**emantic-**C**onsistent Diffusion framework (FedDISC)[3], which innovatively integrates federated learning with generative modality recovery to address challenges posed by incomplete modalities. To alleviate the dependency of generative modality recovery on modality completeness, FedDISC establishes a federated architecture where clients train local modality-specific diffusion models using their local available modalities. These models are then aggregated into global modality-specific diffusion models on the server and broadcast to clients missing corresponding modalities. This eliminates the need for clients to locally train recovery models for missing modalities, overcoming the limitations of single-client incomplete modalities while enabling zero-shot cross-client modality recovery and safeguarding data privacy.

Additionally, we design the DISC-Diffusion model, which integrates a Dialogue Graph Network (DGN) to capture contextual dependencies and speaker relationships through graph structures, and a Semantic Conditioning Network (SCN) that extracts semantic information from available modalities via attention mechanisms. The fusion of these two modules ensures tri-dimensional consistency between recovered and available modalities across context, speaker identity, and semantic alignment. Finally, we introduce the Alternating Frozen Strategy (AFS), which cyclically freezes recovery module and classifier module on each client to resolve optimization conflicts between generative and classification objectives during federated collaborative training. The contributions of this work are summarized as follows:

1. **Federated Learning for Modality Recovery:** FedDISC pioneers the integration of federated learning with generative modality recovery, mitigating single-client limitations caused by incomplete modalities while preserving data privacy.

2. **DISC-Diffusion for Consistent Recovery:** We design DISC-Diffusion with DGN and SCN modules to ensure tri-dimensional consistency (context, speaker identity, semantics) between recovered and available modalities, leveraging dialogue and semantic constraints.

3. **Alternating Frozen Strategy:** AFS effectively eliminates cross-modal optimization conflicts by alternately freezing recovery and classifier modules during federated updates.

## 2 Background

### 2.1 Incomplete Multimodal Learning

Incomplete multi-modal learning, a critical research topic in machine learning, aims to enhance models' capability to handle unpredictable and inevitable modality missing in real-world scenarios [17]. It can be categorized into two paradigms based on whether to restore missing modalities: non-recovery methods and recovery methods [18].

Non-recovery methods focus on inferring directly from incomplete modal inputs by improving model architectures or optimization strategies. Representative approaches include knowledge distillation-based and correlation maximization techniques. Knowledge distillation methods employ teacher networks to learn modality-specific predictive models, then distill knowledge from available modalities to student networks [19, 20]. Correlation maximization methods aim to maximize cross-modal dependencies and enforce shared low-dimensional embeddings across heterogeneous modalities through covariance constraints or mutual information maximization [21, 22].

---

[3]Code Repository: `https://github.com/wdqdp/FedDISC`.

Recovery methods aim to estimate and reconstruct missing modalities from available modalities. It can be categorized into latent space semantic recovery and explicit modality recovery. The former recovers missing semantics in latent spaces by mining deep dependencies from multimodal data. Lian et al. [12] uses graph networks to capture temporal and speaker dependencies in dialogues, enabling semantic compensation for incomplete modalities within an invisible latent space. Zhang et al. [23] focuses on processing high-frequency signals via graph neural networks, achieving more comprehensive modality recovery through multi-frequency feature alignment compared with [12]. Explicit modality recovery usually employs generative models to directly reconstruct missing modalities. For example: Wang et al. [18] utilizes normalizing flow-based generative model DiCMoR to predict missing modality distributions, exploiting the reversible property and precise density estimation of flow models to ensure distributional consistency. Liu et al. [9] proposes CIF-MMIN, which integrates an invariant feature-guided imagination module and cascaded residual autoencoders to generate missing modality features that maintain semantic coherence with available modalities. However, the inherent dependency of existing methods on data completeness fundamentally restricts their practical effectiveness in real-world scenarios with severe modality missing.

## 2.2 Incomplete Multimodal Federated Learning

Multimodal federated learning(MFL) enables decentralized cross-modal learning while preserving data privacy. However, the inevitable presence of missing modalities in real-world scenarios poses significant challenges to its effectiveness. Le et al. [24] propose Multimodal Federated Cross Prototype Learning (MFCPL) to addresses the issue of missing modalities through prototype learning. They introduce Cross-Modal Alignment (CMA) to align zero-padded features of missing modalities with existing modality features, reducing noise from zero-padding. However, this method can only deeply mine existing data at the prototype level, failing to fundamentally reconstruct missing information. To tackle this, Yin et. al. [25] introduce Stable Diffusion into MFL to recover missing modalities. On image-text datasets with missing image modalities, clients upload text embeddings, and the server generates and extracts image modality features. However, this method deploys a global generative model on the server, unable to generate client-specific local feature image modalities, limiting its generalization. To address these issues, our FedDISC framework fine-tunes modality-specific feature recovery modules for each client's local characteristics.

## 3 Method

### 3.1 Dialogue-guided and Semantic-Consistent Module

This chapter focuses on generative modality recovery methods under fixed missing protocol scenarios. To ensure consistency between the recovered modality and the available modalities across context, speaker identity, and semantic alignment, FedDISC first pretrain two modules, DGN and SCN, as illustrated in Figure 2. We define a conversation has $\mathcal{C}$ utterances: $C = \{u_i\}_{i=1}^{\mathcal{C}}$, and each utterance $u_i$ is spoken by speaker $p_{s(u_i)}$, where $s(\cdot)$ is a map between the utterance and its speaker. In fixed missing protocol, each client retains data from at least one modality, resulting in a total of $(2^M - 1)$ possible missing patterns, where $M$ is the number of all modalities. Here, we consider three modalities: *language* (*l*), *vision* (*v*), and *acoustic* (*a*), and we will discuss the case of missing *l* modality.

### 3.1.1 Dialogue Graph Network

The core idea of our DGN is to capture context and speaker dependencies from conversational utterances, which have been proven to be essential for dialogue understanding [26, 27]. Due to the inconsistent dimensional spaces across different modalities, we first employ a feature extractor composed of pre-trained embedding layers and convolution layers to project the raw data into a unified dimensional space $H = \{h_i^v, h_i^a\}_{i=1}^{\mathcal{C}}$. We construct separate speaker graph and context graph for each available modality to extract modality-aware dialogue information.

Taking modality *a* as an example: $H_a = \{h_i^a\}_{i=1}^{\mathcal{C}}$ is utilized as the initial nodes. In graphs, edge weights quantify the importance of connections between nodes, while edge types reflect their interaction patterns. The context graph and speaker graph share the same graph structure, but employ distinct edge types to capture different dependencies. To avoid excessive computational

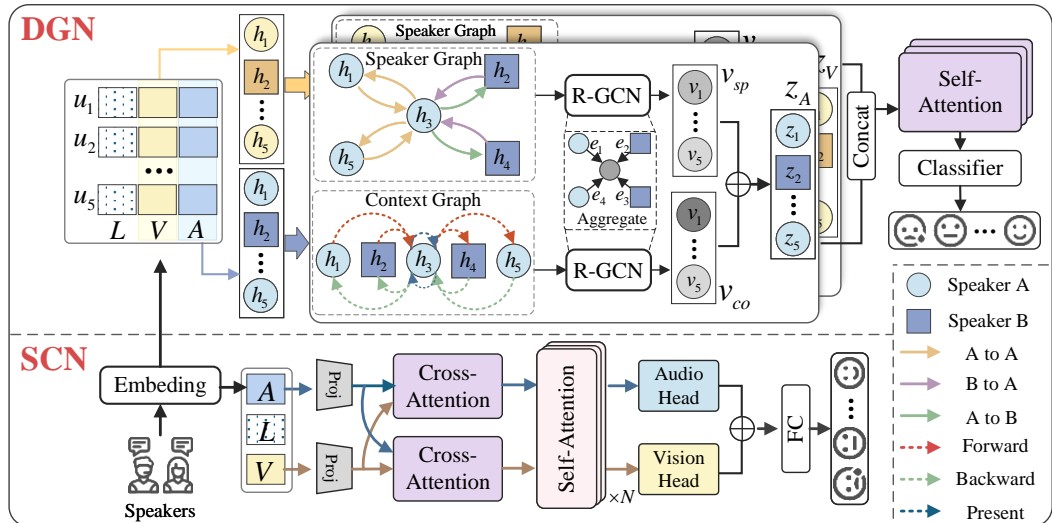

Figure 1: The frame work of DGN and SCN. DGN captures context and speaker dependencies through graph network while SCN captures cross-modal semantic information with attention mechanism.

overhead from connecting all nodes, we constrain the graph scale by implementing a fixed-size window $w$. We select $w$ from the set $\{1, 2, 3\}$, and a node $h_i$ can only connect within the node set $\{h_j\}_{j \in [max(i-w,1), min(i+w,\mathcal{C})]}$. Then we can denote the edge from $h_i$ to $h_j$ as $e_{ij} \in \mathcal{E}_i$. We set $w = 2$ in Figure 1.

**Speaker Graph**: This graph leverages speaker relation to capture speaker-aware dependencies in conversations. Let $\alpha_{ij} \in \mathcal{A}$ denote the speaker relation identifier of $e_{ij}$, the value of $\alpha_{ij}$ reflects the flow from $p_{s(u_i)}$ to $p_{s(u_j)}$. Therefore, $|\mathcal{A}| = 3$ when the conversation happens between two speakers: $\{i \rightarrow j, j \rightarrow i, i \rightarrow i\}$.

**Context Graph**: This graph leverages context relation to capture context-aware dependencies in conversations. Let $\beta_{ij} \in \mathcal{B}$ denote the context relation identifier of $e_{ij}$, the value of $\beta_{ij}$ can be categorized as {forward, present, backward} based on the relative positions of node $h_i$ and $h_j$ in the conversation.

**Graph learning** We employ single-layer R-GCN [28] to aggregate speaker and context information under multiple relations. The calculation formula is shown as follows:

$$v_i^s = \sigma\Big(\sum_{r \in \alpha} \sum_{j \in N_i^r} \frac{1}{|N_i^r|} W_r^s h_j\Big), \ v_i^c = \sigma\Big(\sum_{r \in \beta} \sum_{j \in N_i^r} \frac{1}{|N_i^r|} W_r^c h_j\Big), \tag{1}$$

where $v_i^s$ and $v_i^t$ are the output of Speaker Graph and Context Graph, respectively. $N_i^r$ is the set of connected indexes of $h_i$ under relation $r$, $W_r$ is the weighted parameters for different types of graph under relation $r$. $\sigma(\cdot)$ is the activation function, we leverage *ReLU* function in this paper. After the aggregation, we add corresponding nodes together to get the dialogue graph representation for each available modality: $z_i^m = v_i^{sm} + v_i^{cm}, m \in \{a, v\}$. To predict the category of each utterance, we concatenate modality-specific graph representations from available modalities, which are then processed through multi-head self-attention layers and a multi-layer perceptron (MLP) to yield classification probabilities: $\hat{y}_d \in \mathbb{R}^{\mathcal{C} \times c}$, $c$ is the number of emotion classes. Then we can get the loss of DGN with cross-entropy loss for pretraining: $\mathcal{L}_{DGN} = -\frac{1}{\mathcal{C}} \sum_{i=1}^{\mathcal{C}} y_i log(\hat{y}_d)$, where $y_i$ denotes the ground truth of utterances.

### 3.1.2 Semantic Conditioning Network

We propose the SCN to capture cross-modal semantic information from available modalities for semantic alignment. Similar to DGN, we utilize pre-trained embedding layers and projection layers to transform raw data from different modalities into a dimensionally aligned feature space: $H = \{h_i^v, h_i^a\}_{i=1}^{\mathcal{C}}, h \in \mathbb{R}^{\mathcal{C} \times d}$, d is the shared dimension.

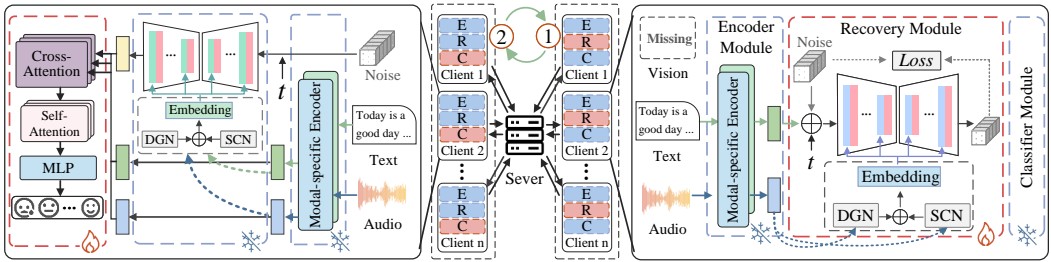

Figure 2: The training pipeline of FedDISC, this figure delineates a hierarchical federated learning framework with two alternating phases: the Recovery Module Training Stage (right) and the Classifier Optimization Stage (left).

To capture inter-modal dependencies, we employ cross-attention layers between audio and visual features. For modality pair $(h^a, h^v)$, the cross-attention mechanism calculates:

$$z^{a \to v} = Norm(h^a + CrossAttn(h^a, h^v)), \; z^{v \to a} = Norm(h^v + CrossAttn(h^v, h^a)), \quad (2)$$

where $CrossAttn(Q, K = V) = Softmax(\frac{QK^T}{\sqrt{d}}V)$, and $z^{a \to v}, z^{v \to a} \in \mathbb{R}^{\mathcal{C} \times s \times p}$. Each fused feature undergoes self-attention to refine intra-modal representations $z_{self}^a, z_{self}^v$. Then we select the head token of each sequence as the modality-specific semantic summary: $s^a = z_{self}^a[0], s^v = z_{self}^v[0]$. All the heads are concatenated and fed into a classifier to predict the emotion category $\hat{y}_s$. Consistent with DGN, loss of SCN is calculated as: $\mathcal{L}_{SCN} = -\frac{1}{\mathcal{C}} \sum_{i=1}^{\mathcal{C}} y_i log(\hat{y}_s)$. Finally, we combine these two loss functions into a joint objective function which is utilized to pretrain the Dialogue-guided and Semantic-Consistent Module: $\mathcal{L} = \mathcal{L}_{DGN} + \mathcal{L}_{SCN}$.

## 3.2 Federated Dialogue-Guided and Semantic-Consistent Diffusion

We consider a centralized federated learning architecture similar to FedAvg [29], where a sever coordinates $n$ clients with incomplete local dataset $\mathcal{D}_l = \{x_i^m, y_i\}_{i=1}^{N_l}, m \in M_l^{avail}$, where $N_l$ is the number of samples in the $l^{th}$ client, $M^{avail}$ is the set of available modalities, with $1 \le |M_l^{avail}| < M$.

### 3.2.1 DISC-Diffusion Model

Due to ethical concerns and processing convenience, our DISC-Diffusion model generates latent features for missing modalities rather than raw data.

**Conditional embedding fusion**: We leverage the SGN and SCN, both pretrained on local data, to respectively extract dialogue dependency and semantic alignment information from the available modalities. Let $z^D = concat(z^m), m \in M^{avail}$ denote the concatenation of the graph network outputs corresponding to each available modality in the DGN, and $z^S$ denote the concatenation of the heads corresponding to each available modality in the SCN, where $z^D, z^S \in \mathbb{R}^{N_l \times d}$. The embedding layer concatenates the two representations and projects the result into the conditional embedding space. The formula is:

$$c = Reshape(Linear(z^D || z^S)), c \in \mathbb{R}^{N_l \times s \times p}. \quad (3)$$

**Training process**: Recent work has demonstrated the effectiveness of conditional diffusion models [30]. Inspired by its success, we implement the conditional noise prediction network with a U-Net architecture [31], where cross-attention layers are used to integrate the intermediate feature of the U-Net with the conditioning signal $c$:

$$F^{(l+1)} = Concat(F^{(l)}, CrossAttn(F^{(l)}, Linear(c)), \quad (4)$$

where $F^{(l)}$ is the input of the $l^{th}$ layer of U-Net, $Linear(\cdot)$ projects $c$ to the same size of $F^{(l)}$. To amplify the condition, we randomly omit the condition with a fixed probability $p$ during training to enable unconditional predictions, and during inference we merge the conditional and unconditional outputs with a weighted combination:

$$\hat{\epsilon}_t = (1 + w)\, \epsilon_\theta(z_t, t, c) - w\, \epsilon_\theta(z_t, t), \quad (5)$$

further, to optimize the network, we can obtain the noise matching objective function:

$$\mathcal{L} = \mathbb{E}_{t \sim \mathcal{U}\{1,\ldots,T\} z_0 \sim p(z_0),\, \epsilon \sim \mathcal{N}(0,I)} \Big\| \epsilon - \big((1+w)\, \epsilon_\theta(z_t, t, c) - w\, \epsilon_\theta(z_t, t)\big) \Big\|^2. \tag{6}$$

**Sampling process**: During the reverse denoising process, we guide the noise prediction with the conditional embedding and define the reverse transition distribution as:

$$p_\theta\big(z_{t-1} \mid z_t, c\big) = \mathcal{N}\big(z_{t-1};\, \tilde{\mu}_t(z_t, t, c),\, \tilde{\beta}_t I\big), \tag{7}$$

where $\tilde{\mu}_t(z_t, t, c)$ is calculated as:

$$\tilde{\mu}_t(z_t, t, c) = \frac{1}{\sqrt{\alpha_t}} \Big( z_t - \frac{1-\alpha_t}{\sqrt{1-\bar{\alpha}_t}} \epsilon_\theta(z_t, t, c) \Big), \tag{8}$$

where $\alpha, \bar{\alpha}$ denote the noise scheduling coefficient. Then sampling is performed using the reparameterization trick.

### 3.2.2 Alternating Frozen Strategy

As illustrated in Figure 2, to harmonize the optimization of recovery and classification modules, we propose the AFS, a two-stage, hierarchical federated learning protocol that systematically freezes and activate model components. Each client's local model is partitioned into three modules: 1) *Encoder Module*: a pretrained feature extractor, whose parameters remain fixed throughout all training; 2) *Recovery Module*: employs DISC-Diffusion for modality recovery; 3) *Classifier Module*: a downstream predictor that is composed of attention layers and a MLP.

**Stage I**: In the first stage, as the right part of Figure 2 shows, only the Recovery Module is active, while the Classifier Module is frozen. This stage has three steps: 1) *Local Update*: Client $l$ trains its local modality-specific DISC-Diffusion model $\theta_l^m$ using locally available modality $m$, where cross-modal data $\mathcal{D}_l^{C_l^m}, C_l^m \in M_l^{avail}$ serves as the conditioning input according to equation 6. 2) *Server Aggregation*: After $e$ local epochs, each client uploads its modality-specific DISC-Diffusion model to the server. The server then performs modality-specific aggregation following equation **??**, yielding three global diffusion models $\{\theta_g^m\}, m \in M$ — one per modality. 3) *Broadcast*: These aggregated global models are sent back to all clients for the next stage.

**Stage II**: In the second stage, as the left part of Figure 2 shows, we invert the freezing scheme: the Recovery Module is frozen, and the Classifier Module becomes active. We first apply the global diffusion model $\{\theta_g^m\}, m \in M^{miss}$ to recover the feature of the missing modality $m$ after equation 8, thereby obtain a complete-modality representation. The recovered full-modal features pass through a cross-attention layer, a self-attention layer, and an MLP to produce the output $\hat{y} \in \mathbb{R}^{N_l \times c}$. The loss function for classifier optimization is $\mathcal{L} = -\frac{1}{N_l} \sum_{i=1}^{N_l} y_i log(\hat{y}_i)$. Then we leverage equation **??** to aggregate and update the classifier modules of all clients.

By alternating between these two stages—freezing the classifier during recovery aggregation and freezing recovery during classifier refinement—every $E$ communication rounds, we prevent gradient interference and ensure that each module converges under a coherent training signal. Empirically, this strategy leads to stable optimization and superior generalization for both modality recovery and emotion recognition.

## 4 Experiments

### 4.1 Datasets and Implementation Details

**Datasets**: To verify the effectiveness of FedDISC, we conduct experiments on three benchmark conversational datasets: IEMOCAP [32], CMU-MOSI [33], and CMU-MOSEI [34]. IEMOCAP consists of 151 conversations between two speakers. For a fair comparison, we employ two prevalent labeling methods, generating datasets with four classes [35] and six classes [12]. CMU-MOSI consists of 2199 utterances, and CMU-MOSEI contains 22856 utterances.

**Evaluation metrics**: IEMOCAP is labeled in categorical labels, therefore we use 4-class accuracy (ACC4), 6-class accuracy (ACC6), and weighted average F1-score (WAF1) [35] as our evaluation metric. Lables of CMU-MOSI and CMU-MOSEI are scored between $[-3, 3]$. This paper focuses

| Dataset | Available | Baseline | FedDISC (P) | FedDISC (I) | GCNet [12] | CIF-MMIN [38] | SDR-GNN [23] | IMDer [8] | DiCMoR [18] |
|---|---|---|---|---|---|---|---|---|---|
| IEMOCAP4 | {l} | 59.6/59.4 | 68.0/68.2 | 65.4/65.2 | 66.7/65.9 | 58.0/59.3 | 66.2/66.2 | 60.4/60.4 | 28.6/28.5 |
| | {v} | 58.1/57.3 | 62.5/60.3 | 60.1/59.5 | 55.7/55.7 | 53.3/51.3 | 56.3/55.6 | 56.2/54.7 | 21.9/15.6 |
| | {a} | 53.0/50.4 | 61.2/60.6 | 56.2/56.1 | 56.8/56.3 | 56.3/58.4 | 57.8/57.3 | 50.8/50.7 | 34.0/27.2 |
| | {l, v} | 67.2/67.3 | 75.8/75.8 | 73.2/73.0 | 64.9/65.0 | 73.2/74.0 | 68.1/68.0 | 67.3/67.3 | 37.0/25.6 |
| | {l, a} | 66.5/65.8 | 78.0/78.1 | 74.3/73.6 | 68.7/68.3 | 74.3/75.6 | 73.0/72.1 | 65.9/66.2 | 32.5/32.3 |
| | {v, a} | 60.6/60.3 | 68.7/68.0 | 67.4/67.5 | 60.4/60.8 | 65.7/66.9 | 60.2/60.3 | 65.0/64.8 | 47.1/37.8 |
| | {l, v, a} | 78.6/78.4 | 78.6/78.4 | 78.6/78.4 | 78.4/78.3 | 78.3/78.5 | 78.5/78.1 | 78.1/78.3 | 78.2/78.3 |
| IEMOCAP6 | {l} | 46.4/45.8 | 56.8/56.5 | 53.6/53.6 | 50.8/50.0 | 54.7/54.3 | 58.8/58.9 | 44.6/44.9 | 34.5/33.8 |
| | {v} | 36.2/36.4 | 56.0/55.5 | 46.3/46.3 | 55.7/55.7 | 39.7/35.2 | 41.8/41.0 | 39.3/35.1 | 33.6/32.4 |
| | {a} | 36.8/29.9 | 60.6/59.9 | 52.2/51.6 | 56.8/56.3 | 56.3/58.4 | 51.6/50.7 | 39.2/37.5 | 38.6/38.7 |
| | {l, v} | 49.4/49.9 | 57.4/57.6 | 54.8/55.0 | 49.3/47.8 | 51.3/52.1 | 60.6/60.3 | 49.6/49.0 | 34.4/34.8 |
| | {l, a} | 51.3/50.6 | 59.4/59.3 | 58.1/58.0 | 51.9/51.3 | 53.8/50.1 | 60.3/60.4 | 51.5/50.4 | 37.1/37.0 |
| | {v, a} | 41.1/36.3 | 66.4/66.3 | 52.8/52.9 | 44.0/43.4 | 56.5/56.9 | 60.0/50.7 | 43.8/41.2 | 36.5/35.9 |
| | {l, v, a} | 64.3/64.7 | 64.3/64.7 | 64.3/64.7 | 58.6/59.1 | 61.3/60.2 | 61.3/61.2 | 58.7/58.4 | 55.9/56.2 |

Table 1: Comparison results on IEMOCAP4 and IEMOCAP6 datasets across different available modalities. The best result in each row is highlighted in dark green.

on the negative/positive classification task, scores less than 0 are mapped to negative and greater than 0 are mapped to positive. For both datasets, we choose the accuracy (ACC) and the WAF1 as evaluation metric [36, 37].

**Implementation details**: In line with previous works [11, 8], We tested different models on two commonly used protocols: 1) random missing protocol, 2) fixed missing protocol. For random missing protocal, we define the missing rate as $\eta = 1 - \frac{\sum_{i=1}^{N_l} m_i}{N_l \times M}$, where $m_i$ denotes the number of available modalities for the $i^{th}$ sample, $N_l$ indicates the sample number of client $l$. At the same time, we ensure that at least one modality is available for each sample, s.t. $m_i \geq 1$ and $\eta \leq \frac{M-1}{M}$. In this paper, we set $M$ to 3, so we have $\eta \in [0.0, 0.1, \cdots, 0.7]$. For fixed missing protocol, We define that each client completely lacks $n$ modalities. To ensure at least one modality is retained, we set $1 \leq n < 3$. Consequently, the possible modality absence patterns for each client can be enumerated as: $(\{l\}, \{v\}, \{a\}, \{l, v\}, \{l, a\}, \{v, a\})$. For federated learning, we set the number of clients as $n_c = 3$, each client evenly and randomly allocated all training, validation, and test data. During the training process, we set the local epoch $e$ to 1, and the communication round $E$ to 3, the window size $w = 2$. We perform five-fold cross-validation [12] and report the mean values on the test set. All experiments were conducted on two NVIDIA L40S GPUs, each equipped with 48 GB of memory.

| Dataset | Method | Missing rate | | | | | | | | Average |
|---|---|---|---|---|---|---|---|---|---|---|
| | | 0.0 | 0.1 | 0.2 | 0.3 | 0.4 | 0.5 | 0.6 | 0.7 | |
| MOSI | FedDISC (P) | 86.3/86.3 | 85.9/85.8 | 83.3/83.2 | 81.1/81.0 | 79.7/79.6 | 80.2/80.2 | 77.5/77.5 | 75.7/75.8 | 81.2/81.2 |
| | FedDISC (I) | 85.9/85.9 | 84.6/84.6 | 82.8/82.9 | 80.6/80.6 | 79.3/79.3 | 77.5/77.6 | 70.5/70.4 | 70.9/70.4 | 79.0/78.9 |
| | GCNet[†] [12] | 85.1/85.2 | 82.3/82.3 | 79.5/79.4 | 77.2/77.2 | 74.4/74.3 | 69.8/70.0 | 66.7/67.7 | 65.4/65.7 | 75.1/75.2 |
| | CIF-MMIN [†] [38] | 84.0/83.6 | 82.5/84.1 | 82.6/82.7 | 80.5/80.9 | 78.4/78.1 | 77.3/77.2.2 | 74.1/74.29 | 69.8/71.3 | 78.8/78.9 |
| | SDR-GNN[†] [23] | 86.3/86.3 | 85.0/85.1 | 81.9/81.9 | 80.7/80.8 | 77.9/78.0 | 76.1/76.2 | 72.2/72.2 | 71.1/71.3 | 78.9/79.0 |
| | IMDER[†] [8] | 85.7/85.6 | 84.9/84.8 | 83.5/83.4 | 81.2/81.0 | 78.6/78.5 | 76.2/75.9 | 74.7/74.0 | 71.9/71.2 | 79.6/79.3 |
| | DiCMor[†] [18] | 85.6/85.7 | 83.9/83.9 | 82.0/82.1 | 80.2/80.4 | 77.7/77.9 | 76.4/76.7 | 73.0/73.3 | 70.8/71.1 | 78.7/78.9 |
| | DCCAE[†] [39] | 77.3/77.4 | 74.5/74.7 | 71.8/71.9 | 67.0/66.7 | 63.6/62.8 | 62.0/61.3 | 59.6/58.5 | 58.1/57.4 | 66.7/66.3 |
| MOSEI | FedDISC (P) | 86.8/86.4 | 86.5/68.1 | 86.4/86.3 | 85.6/84.7 | 84.5/84.3 | 82.6/83.2 | 81.7/82.5 | 81.5/82.2 | 84.4/84.5 |
| | FedDISC (I) | 85.4/85.2 | 84.8/84.7 | 84.0/84.3 | 83.4/82.3 | 82.3/82.9 | 81.9/80.1 | 80.2/81.2 | 80.5/81.0 | 82.8/82.8 |
| | GCNet[†] [12] | 85.1/85.2 | 82.1/82.3 | 79.9/80.3 | 76.8/77.5 | 74.9/76.0 | 73.2/74.9 | 72.1/74.1 | 70.4/73.2 | 76.8/77.9 |
| | CIF-MMIN [†] [38] | 85.8/86.2 | 85.4/85.5 | 85.0/85.3 | 83.1/83.8 | 82.7/82.5 | 80.4/81.1 | 78.5/79.2 | 77.3/77.4 | 82.3/82.6 |
| | SDR-GNN[†] [23] | 87.3/87.4 | 86.7/86.8 | 85.7/85.9 | 84.7/84.8 | 83.8/84.0 | 82.6/82.8 | 81.3/81.6 | 80.8/81.0 | 84.1/84.3 |
| | IMDER[†] [8] | 85.1/85.1 | 84.8/84.6 | 82.7/82.4 | 81.3/80.7 | 79.3/78.1 | 79.0/77.4 | 78.0/75.5 | 77.3/74.6 | 80.9/79.8 |
| | DiCMor[†] [18] | 85.1/85.1 | 83.5/83.7 | 81.5/81.8 | 79.3/79.8 | 77.4/78.7 | 75.8/77.7 | 73.7/76.7 | 72.2/75.4 | 78.6/79.9 |
| | DCCAE[†] [39] | 81.2/81.2 | 78.3/78.4 | 75.4/75.5 | 72.2/72.3 | 70.0/70.3 | 66.4/69.2 | 63.2/67.6 | 62.6/66.6 | 71.2/72.6 |

Table 2: Comparison results on CMUMOSI and CMUMOSEI datasets under different missing rates. [†] indicates the results come from [23].

## 4.2 Comparison with SOTA Methods

We compare our FedDISC approach with state-of-the-art recovery methods under unified environmental and dataset settings: GCNet [12], MMIN [40], SDR-GNN [23], IMDer [8], and DiCMoR [18]. Also, we consider one non-recovery method with classical correlation maximization DCCAE [39] to make a comprehensive comparison. To evaluate the generalization of our proposed framework,

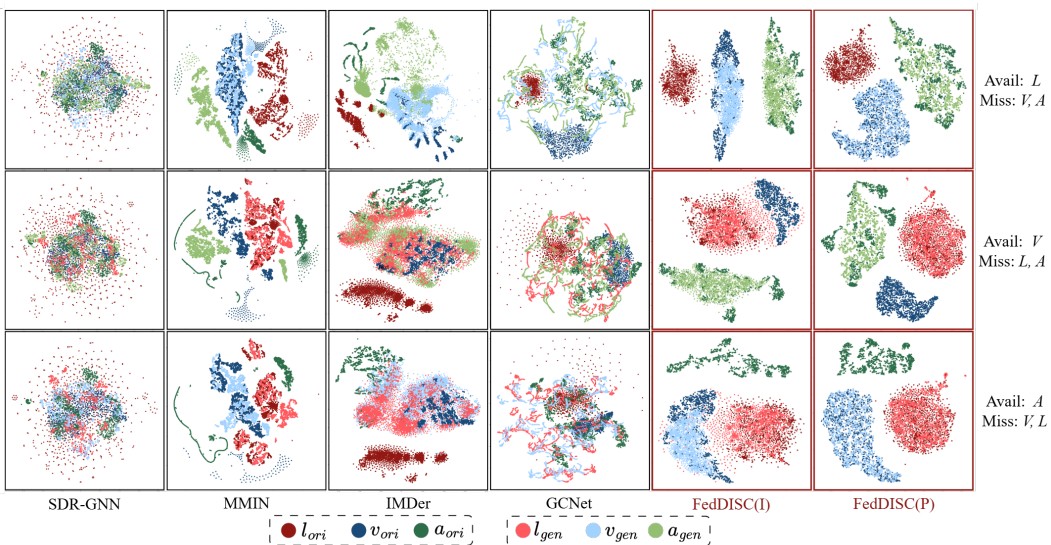

Figure 3: The t-SNE visualization compares the modality recovery performance of different methods under single-modality availability. the features generated by FedDISC exhibit higher distributional similarity to the original modality features compared to other methods, demonstrating its effectiveness.

we develop DISC-Diffusion with two classical diffusion models: DDPM (timesteps $t = 1000$) and DDIM (timesteps $t = 50$), denoted as FedDISC (P) and FedDISC (I) separately. The results are reported below.

**Fixed missing protocol**. As listed in Table 1, our proposed FedDISC (P) achieves the highest accuracy and WAF1 under nearly all partial modality incomplete conditions. Specifically, the FedDISC (P) delivers optimal classification results across all scenarios on the IEMOCAP4 dataset, outperforming the best-performing SOTA methods by $1.3\% \sim 7.5\%$ in accuracy. On IEMOCAP6, the FedDISC (P) slightly lags behind the SDR-GNN by $0.9\% \sim 3.2\%$ in certain incomplete modality settings ($\{l\}$, $\{l, v\}$, $\{l, a\}$). However, considering federated clients are assigned only $N_l = \frac{\mathcal{C}}{n_c}$ samples, while other SOTA methods are full training data accessible, this result highlights our method's capability to generate high-quality missing modality features under few-shot environments. Wilcoxon signed-rank tests with the alternative hypothesis set to "greater" show $W = 21.00, p = 0.0156 < 0.05$, indicating significant superiority of FedDISC(P). The effect size, computed as rank-biserial correlation $r$ ($r = Z/\sqrt{N}, N = 6$), is $r \approx 0.77$, reflecting a large effect ($r > 0.5$).

**Random missing protocol**. Table 2 illustrates the comparison results under missing rates range from 0.0 to 0.7. On the CMUMOSI and CMUMOSEI datasets, our FedDISC exhibits slightly lower accuracy and WAF1 compared to SOTA models when the missing rate lower than 0.4. However, it achieves the highest accuracy and WAF1 under missing rates $\geq 0.4$. This discrepancy arises because,

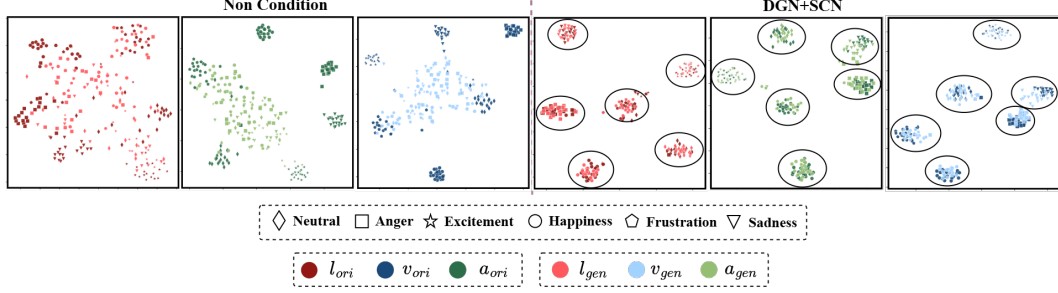

Figure 4: The visualized ablation study on the IEMPCAP6 dataset. Compared with unconditional modality recovery, DGN and SCN guide recovery by leveraging dialog and semantic alignment, ensuring category consistency between recovered and original modalities.

with low missing rates (e.g., $< 0.4$), modality recovery models are of low importance in classification tasks. While as the missing rate increases, classification performance becomes more dependent on modality recovery capabilities. As shown in the table, FedDISC is less suffered by rising missing rates than other models. Specifically, when the missing rate increases from 0.0 to 0.7, on CMU-MOSI, the accuracy of the compared models declines about $15.4\% \sim 27.7\%$ while our FedDISC shows only a $12.3\%$ decline. on CMU-MOSEI, the compared models declines by $7.4\% \sim 22.9\%$ while FedDISC declines merely $6.1\%$. These comparisons validate the generalization capability of FedDISC's recovery model under random missing protocol. Wilcoxon tests yield $W = 45.00, p = 0.0020 < 0.05$, confirming significant differences. The effect size is $r \approx 0.81$ ($N = 8$), also indicating a large effect.

**Visualization experiments**. Figure 3 illustrates the distribution plots of missing modality features generated by our FedDISC and other recovery-based models under the fixed missing protocol with single modality availability, compared to ground truth features. We conducted experiments on the IEMOCAP's test set from a single client, projecting these features into a 2D space via t-SNE. Observations reveal that FedDISC (P) achieves the highest denoising capability, predicting features that closely align with the original distribution. FedDISC (I) slightly underperforms FedDISC (P) in recovering modalities $l$ and $v$, yet still surpasses other recovery methods. This demonstrates FedDISC's effectiveness in ensuring distributional consistency between recovered and original data.

| Available | $l$ | | $v$ | | $a$ | | $l, v$ | | $l, a$ | | $v, a$ | |
|---|---|---|---|---|---|---|---|---|---|---|---|---|
| AFS | ACC4 | WAF1 | ACC4 | WAF1 | ACC4 | WAF1 | ACC4 | WAF1 | ACC4 | WAF1 | ACC4 | WAF1 |
| ✗ | 52.3 | 50.8 | 46.8 | 47.1 | 50.2 | 50.5 | 61.5 | 61.2 | 59.7 | 58.9 | 66.0 | 65.8 |
| ✓ | 68.0 | 68.2 | 62.5 | 60.3 | 61.2 | 60.6 | 75.8 | 75.8 | 78.0 | 78.1 | 68.7 | 68.7 |

Table 3: Ablation study under different modality-missing scenarios on the IEMOCAP4 dataset. This table shows that AFS enables collaborative training between the recovery module and the classification module under any modality-missing conditions.

## 4.3 Ablation Study

**DGN** & **SCN**. To validate the importance of dialogue and semantic dependencies extracted by DGN and SCN in the proposed DISC-Diffusion model, we design a visualized ablation study. As shown in Figure 4, we visualize the recovered and original modality features of IEMPCAP6 under two conditions: 1) Unconditional recovery (no guidance), 2) Jointly conditioned on DGN and SCN. Specifically, we randomly select 120 samples (20 per class) from the IEMOCAP6 test set and project both generated and original features into a 2D space via t-SNE. We can observe that under unconditional recovery setting, recovered features suffer from semantic ambiguity because of the lack of semantic guidance. However, the modalities recovered through conditional recovery guided by DGN and SCN exhibit strong clustering with the original modalities of the same category, while maintaining distinct separation between classes. This validates the effectiveness of DGN and SCN in capturing comprehensive dialogue and semantic information.

**AFS**. We design ablation experiments to validate the effectiveness of the proposed AFS. On the IEMOCAP4 dataset, we test the AFS-equipped model and a baseline model (where the recovery and classification modules are co-trained without AFS) under various modality-missing configurations. Table 3 shows that the model with AFS outperforms the baseline in all missing-modality scenarios. This demonstrates that AFS effectively mitigates optimization conflicts between modules and improves training efficiency.

## 4.4 Computation and Communication Cost

**Computation cost**. The primary source of computational overhead in FedDISC lies in the inference time of the diffusion-based recovery module. In our experiments, using DDPM ($t = 1000$), the average inference time is approximately $2.13 \ min/epoch$; for DDIM ($t = 50$), this reduces significantly to $5.7 \ s/epoch$. As a result, the total training time varies with the diffusion backbone. Importantly, our feature recovery module is not restricted to diffusion models—any conditional generative model with lower inference latency (e.g., conditional autoencoders or GANs) can be flexibly adopted within our framework.

**Communication cost**. The communication cost per client is $10.57\ MB/round$ when using the conditional DDPM. Compared to the traditional federated learning framework, FedAvg, which incurs a communication cost of $22.64\ MB/round$, our proposed hierarchical federated framework successfully reduces the communication overhead by more than half. Notably, the diffusion model parameters account for the majority of this cost—approximately $85\%$, and our framework allows for any lighter-weight conditional generative model to be used as the recovery module. For example, when using DDIM, the communication cost further decreases to $7.18\ MB/round$. Moreover, the recent paper [41] reports that communication costs of $5-12\ MB$ per round per client are fully deployable in real-world scenarios. This strongly supports the practicality and deployability of our approach.

## 5  Conclusion and Limitations

In this work, we try to challenge MERC under extreme incomplete multimodalities by introducing a new modality recovery paradigm: FedDISC. We pioneers the integration of federated learning with generative modality recovery to mitigate the dependence of recovery models on modality integrity. We design DISC-Diffusion with a DGN and a SCN module to capture dialogue and semantic dependencies separately for diffusion guidance. Additionally, the designed AFA strategy balances collaborative training between recovery models and classifiers via a periodic freezing optimization mechanism. Comparative experiments and ablation studies validate the effectiveness of FedDISC in missing-modality recovery and collaborative training.

**Limitations**: First, its sampling strategy can't balance recovery quality and computational cost (e.g., iterative denoising requires multi-step inference). Future work may adopt consistency models for faster high-quality generation. Moreover, FedDISC assumes synchronous federated training, while real-world scenarios often involve asynchronous devices and dynamic modality absence. Extending it to asynchronous FL frameworks will enhance practicality.

## Acknowledgements

This research was supported by Guangdong Basic and Applied Basic Research Foundation (No. 2024A1515011774), the National Key Research and Development Program of China (No. 2022YFC3310300), the National Natural Science Foundation of China (No. 12171036), Shenzhen Sci-Tech Fund (Grant No. RCJC20231211090030059).

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

# A  Theoretical Analysis

**Assumption 1 (Noise-Prediction Error Model)** *For all clients $k$ and timesteps $t$,*
$$\mathbb{E}_{z_0,\epsilon}\big[\epsilon - \epsilon_{\theta_k}(z_t, t, c)\big] = 0, \mathbb{E}_{z_0,\epsilon}\big[\|\epsilon - \epsilon_{\theta_k}(z_t, t, c)\|^2\big] \leq \varepsilon_{\text{diff}}^2. \tag{9}$$

**Assumption 2 (Conditioning Embedding Lipschitznes)** *For the dialogue graph encoder $E_{\text{DGN}}$ and semantic encoder $E_{\text{SCN}}$, we require*
$$\begin{aligned}
\big\|E_{\text{DGN}}(x) - E_{\text{DGN}}(x')\big\| &\leq L_{\text{DGN}} \|x - x'\|, \\
\big\|E_{\text{SCN}}(x) - E_{\text{SCN}}(x')\big\| &\leq L_{\text{SCN}} \|x - x'\|.
\end{aligned} \tag{10}$$
*We then set $L_{\text{tot}} := L_{\text{DGN}} + L_{\text{SCN}}$.*

**Assumption 3 (Aggregated Parameter Bias)** *FedAvg after one communication round yields*
$$\|\theta_g - \theta^*\| \leq \delta_{\text{agg}}, \|\phi_g - \phi^*\| \leq \delta_{\text{agg}}, \tag{11}$$
*where $\{\theta^*, \phi^*\}$ minimises $F(\theta) + G(\phi)$ on the pooled (non-federated) data.*

**Assumption 4 (Smoothness + Polyak–Łojasiewicz)** *Both blocks satisfy*
$$\begin{aligned}
\big\|\nabla F(\theta) - \nabla F(\theta')\big\| \leq M \|\theta - \theta'\|, \tfrac{1}{2}\|\nabla F(\theta)\|^2 &\geq \mu\big[F(\theta) - F(\theta^*)\big], \\
\big\|\nabla G(\phi) - \nabla G(\phi')\big\| \leq M \|\phi - \phi'\|, \tfrac{1}{2}\|\nabla G(\phi)\|^2 &\geq \mu\big[G(\phi) - G(\phi^*)\big].
\end{aligned} \tag{12}$$

**Assumption 5 (Bounded SGD Variance)** *For any client gradient estimator $g$,*
$$\mathbb{E}[g] = \nabla\ell, \mathbb{E}\big\|g - \nabla\ell\big\|^2 \leq \sigma^2. \tag{13}$$

**Theorem 1 (Convergence of Recovery-Only Rounds)** *Run FEDDISC for $T$ global rounds with step-size $\eta \leq 1/M$, $E$ local steps, $K$ clients chosen i.i.d., then*
$$\mathbb{E}\big[F(\theta_g^T)\big] - F(\theta^*) \leq \frac{M\|\theta_g^0 - \theta^*\|^2}{2\eta T} + \frac{\eta E \sigma^2}{K \mu} + M \delta_{\text{agg}}^2 + \frac{\varepsilon_{\text{diff}}^2}{\mu}. \tag{14}$$

*Proof.* **One client, one step.** By $M$-smoothness:
$$F(\theta - \eta g) \leq F(\theta) - \eta\langle\nabla F(\theta), g\rangle + \frac{M\eta^2}{2}\|g\|^2. \tag{15}$$

Take expectation over the stochastic gradient and use Assumption 5:
$$\mathbb{E}F(\theta') \leq F(\theta) - \eta\|\nabla F(\theta)\|^2 + \frac{M\eta^2}{2}\big(\|\nabla F(\theta)\|^2 + \sigma^2\big). \tag{16}$$

Since $\eta \leq 1/M$, we get:
$$\mathbb{E}F(\theta') \leq F(\theta) - \frac{\eta}{2}\|\nabla F(\theta)\|^2 + \frac{M\eta^2\sigma^2}{2}. \tag{17}$$

**PL condition Assumption 4.**
$$\|\nabla F(\theta)\|^2 \geq 2\mu\big(F(\theta) - F(\theta^*)\big). \tag{18}$$

Apply Equ. (17) for $E$ local steps:
$$\mathbb{E}[F(\theta_k^t)] \leq F(\theta_g^t) - \eta E\mu \Delta_t + \frac{M\eta^2 E\sigma^2}{2}, \quad \Delta_t := F(\theta_g^t) - F(\theta^*). \tag{19}$$

**FedAvg aggregation + bias Assumption 3.**
$$\Delta_{t+1} \leq (1 - \eta E\mu) \Delta_t + \frac{M\eta^2 E\sigma^2}{2} + M \delta_{\text{agg}}^2. \tag{20}$$

**Solve linear recursion.** Noting $1 - \eta E\mu \leq e^{-\eta E\mu}$ and summing over $t = 0, \ldots, T - 1$,
$$\Delta_T \leq \frac{M\|\theta_g^0 - \theta^*\|^2}{2\eta T} + \frac{\eta E\sigma^2}{K\mu} + M \delta_{\text{agg}}^2. \tag{21}$$

**Add irreducible diffusion gap.** The imperfect predictor incurs an additive floor $\varepsilon_{\text{diff}}^2/\mu$. Combining with equ. (21) yields the final bound. ∎

**Theorem 2 (Error Bound for Recovered Latent Vectors)** *After $T_{\mathrm{rev}}$ reverse diffusion steps, the recovered latent vectors satisfy*

$$\mathbb{E}\big\|\widehat{z}_i - z_i^*\big\| \leq C_{\mathrm{cum}}\, L_{\mathrm{tot}}\, \varepsilon_{\mathrm{diff}} + \eta_{\mathrm{attn}}. \tag{22}$$

*Proof.* From the reverse mean formula and Assumption 1:

$$\|\tilde{\mu}_t - \tilde{\mu}_t^*\| = C_t\, \|\epsilon_\theta - \epsilon\| \leq C_t\, \varepsilon_{\mathrm{diff}}. \tag{23}$$

Conditioning perturbation at step $t$:

$$\|c_t - c_t^*\| \leq L_{\mathrm{tot}}\, C_t\, \varepsilon_{\mathrm{diff}}. \tag{24}$$

Because the U-Net cross-attention [42] is affine in $c_t$, the bound Equ. (24) re-enters additively into the next noise prediction error.

$$\mathbb{E}\big\|\widehat{z}_i - z_i^*\big\| \leq \sum_{t=1}^{T_{\mathrm{rev}}} L_{\mathrm{tot}}\, C_t\, \varepsilon_{\mathrm{diff}} + \eta_{\mathrm{attn}} = C_{\mathrm{cum}}\, L_{\mathrm{tot}}\, \varepsilon_{\mathrm{diff}} + \eta_{\mathrm{attn}}. \tag{25}$$

$\blacksquare$

**Theorem 3 (Linear Rate of Alternating-Freeze Optimisation)** *With alternating blocks $A : \theta$-updates, $B : \phi$-updates, step-sizes $\eta_A, \eta_B \leq 1/M$, and executing $R$ full alternations, we have:*

$$Q(\theta^R, \phi^R) - Q(\theta^*, \phi^*) \leq \big(1 - \tfrac{\mu}{M}\big)^R \big[Q(\theta^0, \phi^0) - Q(\theta^*, \phi^*)\big], \tag{26}$$

*where $Q(\theta, \phi) := F(\theta) + G(\phi)$.*

*Proof.* Define the block operator

$$\mathcal{T} : (\theta, \phi) \mapsto \big(\theta - \eta_A \nabla F(\theta),\ \phi - \eta_B \nabla G(\phi)\big). \tag{27}$$

**One $A$-update with $\phi$ frozen.** By the PL condition for $F$:

$$F(\theta^+) - F(\theta^*) \leq (1 - \eta_A \mu)\big[F(\theta) - F(\theta^*)\big]. \tag{28}$$

**One $B$-update with $\theta$ frozen.** Similarly for $G$,

$$G(\phi^+) - G(\phi^*) \leq (1 - \eta_B \mu)\big[G(\phi) - G(\phi^*)\big]. \tag{29}$$

**Combine.** Let $Q(\theta, \phi) = F(\theta) + G(\phi)$ and define

$$\rho := \max\{1 - \eta_A \mu,\ 1 - \eta_B \mu\} \leq 1 - \frac{\mu}{M}. \tag{30}$$

Then after one full alternation:

$$Q^+ - Q^* \leq \rho\,(Q - Q^*). \tag{31}$$

**After $R$ alternations.** By recursion,

$$Q(\theta^R, \phi^R) - Q^* \leq \rho^R\big[Q(\theta^0, \phi^0) - Q^*\big] = \big(1 - \tfrac{\mu}{M}\big)^R\big[Q(\theta^0, \phi^0) - Q^*\big]. \tag{32}$$

$\blacksquare$

| | |
|---|---|
| $C$ | The set of utterances |
| $\mathcal{C}$ | Number of utterances in a conversation |
| $u_i$ | The $i$-th utterance in a conversation |
| $p_s(u_i)$ | Speaker identity of utterance $u_i$ |
| $H$ | Unified feature space (e.g., $H = \{h_i^m\}$) |
| $h_i^m$ | Feature representation of modality $m$ for utterance $u_i$ |
| $w$ | Fixed window size for graph neighborhood construction |
| $v_i^s, v_i^c$ | Outputs of Speaker Graph and Context Graph for utterance $u_i$ |
| $z_i^m$ | Output of DGN for modality $m$ |
| $\mathcal{D}_l$ | Local dataset of client $l$ |
| $N_l$ | Sample number of client $l$'s dataset |
| $M_l^{\text{avail}}$ | Set of available modalities for client $l$ |
| $z^D, z^S$ | Outputs from DGN and SCN modules |
| $c$ | Conditional embedding for diffusion guidance |
| $F^{(l)}$ | Input features at the $l$-th layer of the U-Net |
| $\epsilon$ | Gaussian noise in the diffusion process |
| $\alpha_t, \bar{\alpha}_t$ | Noise scheduling coefficients at timestep $t$ |
| $\theta_g^m$ | Global diffusion model parameters for modality $m$ |
| $\phi_g$ | Global classifier parameters |
| $E$ | Alternating communication interval of AFS |
| $S_t$ | Subset of clients selected in communication round $t$ |
| $e$ | Number of local training epochs per client |
| $y$ | Lable of the dataset |
| $\hat{y}_d, \hat{y}_s, \hat{y}$ | Predictions of DGN, SCN, and classifier module |

Table 4: Main Notations Used in this Work. This table highlights the key notations and symbols used throughout the paper, providing a concise reference for understanding the mathematical formulations and models presented.

## B Algorithm Analysis

In Algorithm 1, we present the pretraining procedure for the conditional capture networks DGN and SCN.

Algorithm 2 illustrates the training process of FedDISC, detailing how AFS alternately freezes the recovery and classifier modules to enable collaborative optimization.

## C Information Redundancy Analysis

To address potential concerns regarding information redundancy, we further analyze whether the recovered modality features in our framework merely replicate the information contained in the existing modalities. Information redundancy occurs when reconstructed features are highly correlated with, or simply duplicates of, the available modalities, thereby failing to contribute complementary cues for downstream learning. In our design, the DGN and SCN are explicitly introduced to alleviate this issue by jointly modeling local semantic patterns and global dependencies. Consequently, the recovered modalities are not direct copies of the observed ones but context-enriched representations that capture higher-level dependencies. This interpretation is supported by our experimental results in Table 1: if substantial redundancy existed, the performance of the model with recovered modalities would approximate that of the Baseline. However, both FedDISC(P) and FedDISC(I) achieve statistically significant improvements under all missing-modality conditions (one-sided Wilcoxon signed-rank test, $W = 0, p = 0.03125 < 0.05$ on IEMOCAP4 and IEMOCAP6), confirming that the recovered modalities contribute non-redundant information to the overall framework.

## D Data Missing Protocols and Federated Mechanism

As illustrated in Figure 5 (a), missing modality challenges in MERC can be categorized into two scenarios [11]: random missing protocol and fixed missing protocol. Figure 5 (b) shows a brief

**Algorithm 1:** Pre-training Process of DGN & SCN Modules

**Input:** Client datasets $\{\mathcal{D}_l\}_{l=1}^K$, window size $w$, pretraining epochs $P$
**Output:** Pretrained DGN parameters $\Theta_{DGN}$, SCN parameters $\Theta_{SCN}$
Initialize $\Theta_{DGN}$ and $\Theta_{SCN}$;
**for** $epoch = 1, \ldots, P$ **do**
    **for** *each client* $l \in [K]$ *in parallel* **do**
        **DGN Forward: for** *each available modality* $m \in M_l^{avail}$ **do**
            Build speaker graph $\mathcal{G}_s^m$ and context graph $\mathcal{G}_c^m$ with window $w$;
            Perform R-GCN aggregation via Eq.(3) to get $\{v_i^{sm}, v_i^{cm}\}$;
            $z_i^m = v_i^{sm} + v_i^{cm}$;
        **end**
        $\hat{y}_d \leftarrow Attention(Concat(\{z_i^m\}))$;
        **SCN Forward:** $H \leftarrow Encoding(\{\mathcal{D}_l^m\}), m \in M_l^{avail}$;
        Compute cross-attention and self-attention layers to get $\{s^m\}$;
        $\hat{y}_s \leftarrow MLP(Concat(\{s^m\}))$;
        **Joint Optimization:** Calculate $\mathcal{L}_{DGN} = -\frac{1}{|C|} \sum y_i \log \hat{y}_d$;
        Calculate $\mathcal{L}_{SCN} = -\frac{1}{|C|} \sum y_i \log \hat{y}_s$;
        Update $\Theta_{DGN}$ and $\Theta_{SCN}$ via $\nabla(\mathcal{L}_{DGN} + \mathcal{L}_{SCN})$;
    **end**
**end**
**return** $\Theta_{DGN}, \Theta_{SCN}$;

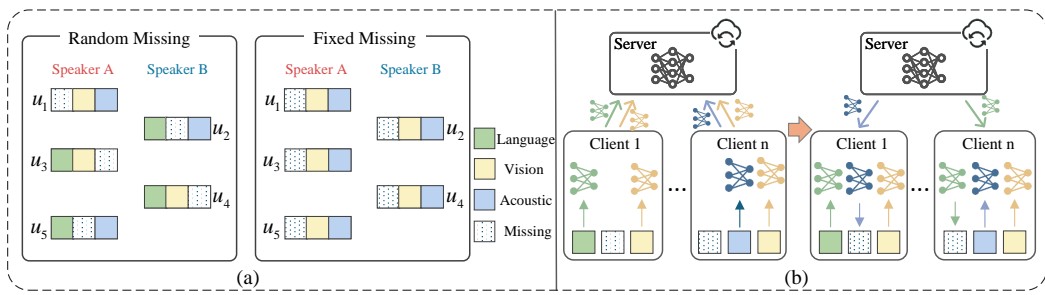

Figure 5: (a) illustrates two types of incomplete modalities: random missing protocol and fixed missing protocol. (b) presents the federated generative modality recovery paradigm proposed in this work, designed to alleviate the dependency of generative recovery on data completeness.

describe of our federated learning mechanism. The clients first train local specialized diffusion models tailored to their available modalities. Then through server-side aggregation, unified modality models are constructed and distributed to modality-deficient clients. By eliminating the need for clients to locally train recovery models for missing modalities, this framework overcomes the limitations caused by single-client incomplete modalities.

## E  Dataset Segmentation and Preprocessing

**Dataset**: As listed in Table 5, IEMOCAP4 includes four types of emotions: anger, happiness (where excitement is merged with happiness), sadness, and neutral [23]. We assign 3290, 1000, and 1241 utterances for train, valid, and test. The six-class dataset IEMOCAP6 encompasses: anger, happiness, sadness, neutral, excitement, and frustration. We assign 4810, 1000, and 1623 utterances for train, valid, and test. CMU-MOSI consists of 2199 utterances, where 1284, 299, 686 samples are set for train, valid, and test. CMU-MOSEI contains 22856 utterances, where 16326 are used for training, 1871 and 4659 samples are used for validation and testing.

**Preprocessing**: For each modality, we employ the corresponding pre-trained network to perform feature extraction. 1) Language: Pre-trained DeBERTa [43] is employed as the language feature extractor. Motivated by its demonstrated superiority in natural language understanding and generation

**Algorithm 2:** Federated Training Process of FedDISC

---

**Input:** Client datasets $\{\mathcal{D}_l\}_{l=1}^K$, global diffusion models $\{\theta_g^m\}_{m \in M}$, alternative interval $E$
**Output:** Global recovery models $\{\theta_g^m\}$, global classifier $\phi_g$
Initialize global models $\theta_l^m \leftarrow \theta_{g_0}^m$, $\phi_l \leftarrow \phi_{g_0}$;
**for** $t = 1, \ldots, T$ *communication rounds* **do**
    **if** $t\%E$ *is odd (Stage I: Recovery Training)* **then**
        **for** *each client* $l \in S_t$ *in parallel* **do**
            **Freeze** classifier module $\phi_l$;
            **for** *each available modality* $m \in M_l^{avail}$ **do**
                Train local recovery model $\theta_l^m$ via Eq.(8) using $\mathcal{D}_l^{C^m}$;
            **end**
            Upload $\{\theta_l^m\}$ to server;
        **end**
        Server aggregates $\theta_g^m \leftarrow \sum_l \frac{|\mathcal{D}_l|}{\sum |\mathcal{D}_i|} \theta_l^m$;
        Broadcast $\{\theta_g^m\}$ to all clients;
    **else if** $t\%E$ *is even (Stage II: Classifier Optimization)* **then**
        **for** *each client* $l \in S_t$ *in parallel* **do**
            **Freeze** recovery modules $\{\theta_g^m\}$;
            Recover missing modalities $\hat{z}_m \leftarrow$ DISC-Diffusion$(\theta_g^{miss}, \mathcal{D}_l^{avail})$;
            Train classifier $\phi_l$ via $\mathcal{L} = -\frac{1}{N_l} \sum y_i \log \hat{y}_i$;
            Upload $\phi_l$ to server;
        **end**
        Server aggregates $\phi_g \leftarrow \sum_l \frac{|\mathcal{D}_l|}{\sum |\mathcal{D}_i|} \phi_l$;
        Broadcast $\phi_g$ to all clients;
    **end**
**end**
**return** $\{\theta_g^m\}, \phi_g$;

---

| Dataset | Train | Val | Test | Total |
|---------|-------|-----|------|-------|
| IEMOCAP4 | 3,290 | 1,000 | 1,241 | 5,531 |
| IEMOCAP6 | 4,810 | 1,000 | 1,623 | 7,433 |
| CMU-MOSI | 1,284 | 229 | 686 | 2,199 |
| CMU-MOSEI | 16,326 | 1,871 | 4,659 | 22,856 |

Table 5: Statistical information on IEMOCAP, CMU-MOSI and CMU-MOSEI.

tasks, we leverage the DeBERTa-large variant to encode utterance sequences into 1024-dimensional representations. 2) Vision: The pre-trained MA-Net [44] serves as the visual feature extractor, utilizing global multi-scale and local attention mechanisms to handle occlusions and non-frontal poses. We first apply MTCNN to detect and align faces, followed by extracting facial features via pre-trained MA-Net. Frame-level features are then compressed into 1024-dimensional utterance-level representations through average encoding. 3) Acoustic: Pre-trained wav2vec [45] serves as the acoustic feature extractor, leveraging its multi-layer convolutional architecture trained on massive unlabeled speech data. Building on its demonstrated success in downstream applications like speech recognition , we adopt wav2vec-large to extract 512-dimensional acoustic features from utterances.

## F  Deep Analysis of Visualization Experiments

Figure 6, as a supplement to Figure 3, illustrates the distribution of features recovered by different modality recovery methods compared to the ground truth, under all modality incomplete conditions. All experiments were conducted on the IEMOCAP dataset. We can observe that, regardless of the modality incomplete condition, the distribution of the features recovered by FedDISC is closer to that of the ground truth compared to SOTA methods.

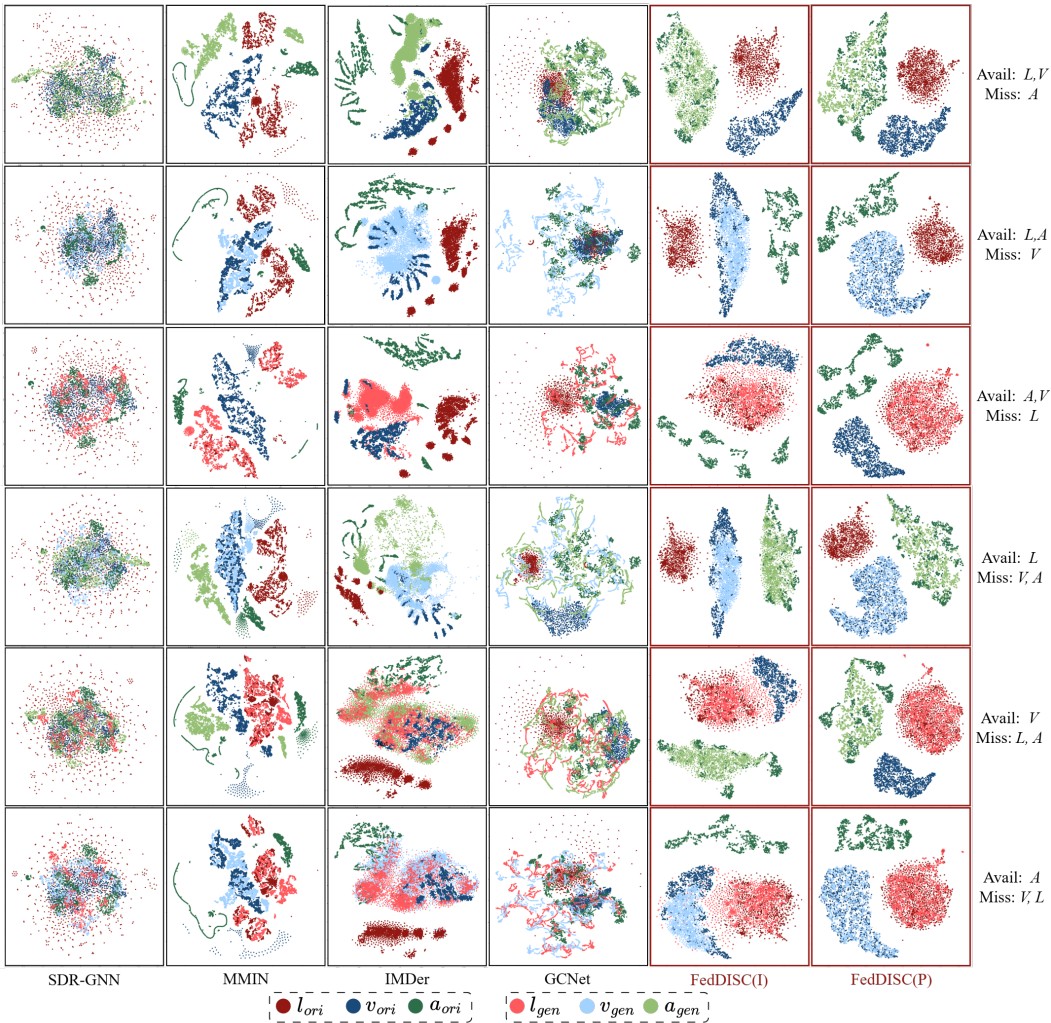

Figure 6: t-SNE visualization: comparison of recovered versus original feature distributions by different modality recovery methods under all modality incomplete conditions on IEMOCAP.

# G  DGN and SCN

Figure 7 demonstrate how the integration of DGN and SCN modules capture dialogue dependencies and semantic information, validating their effectiveness in guiding the conditional diffusion process. In addition to the conclusions observed in sec 4.3, we can further deduce the following: 1) When single module is used, compared to DGN, the semantic information extracted by SCN provides more effective guidance for the modality recovery process. 2) The combination of DGN and SCN modules, as opposed to using a single module, offers more comprehensive dialogue-semantic information, thereby further mitigating semantic confusion in the recovered modalities.

Table 6 extends the aforementioned ablation experiments. We conducted tests on the IEMOCAP4 dataset under six incomplete modality conditions, evaluating four ablation settings and recording their ACC and WAF1, with the highest metrics highlighted in dark green. The table clearly demonstrates that when both SGN and SCN are used as conditional guidance, our model achieves the highest performance across all incomplete modality scenarios. This further validates that the integration of dialogue dependencies and semantic information provides superior guidance for modality recovery.

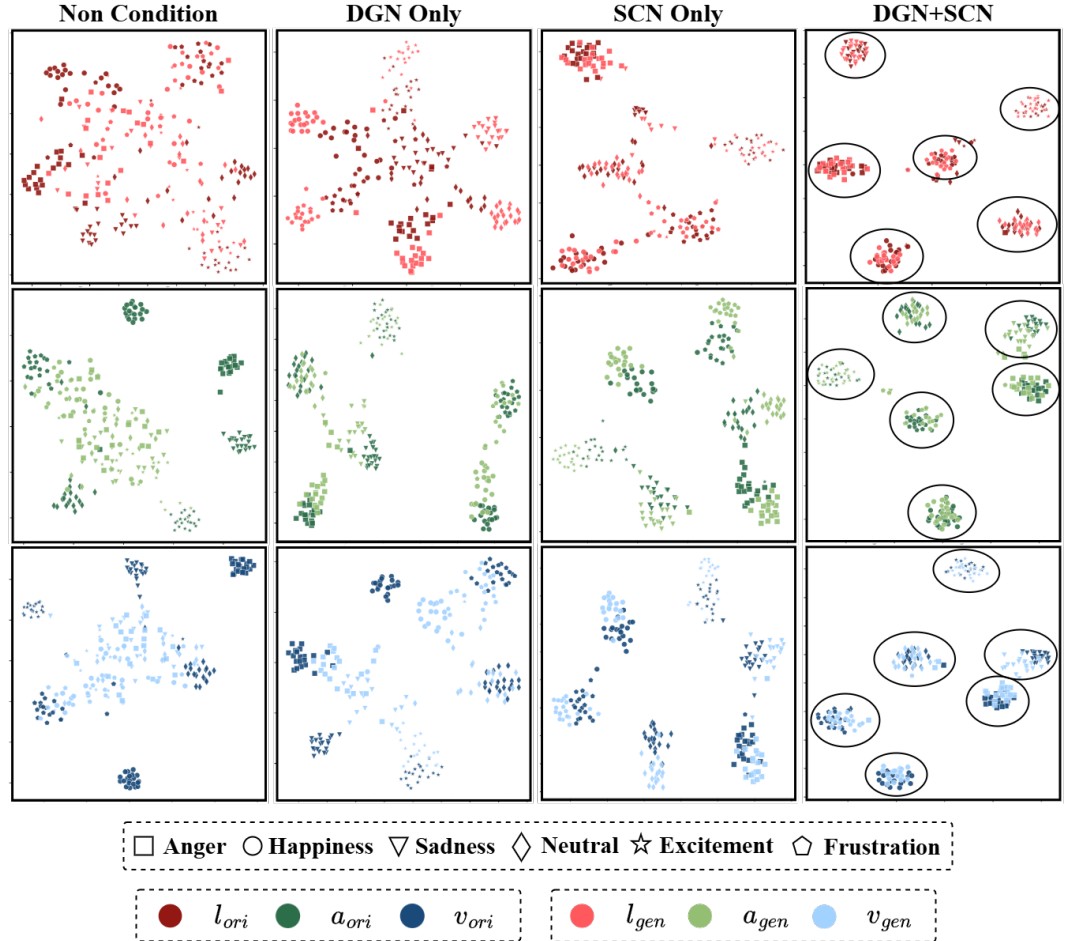

| Non Condition | DGN Only | SCN Only | DGN+SCN |

□ **Anger** ○ **Happiness** ▽ **Sadness** ◇ **Neutral** ☆ **Excitement** ⬠ **Frustration**

● $l_{ori}$   ● $a_{ori}$   ● $v_{ori}$        ● $l_{gen}$   ● $a_{gen}$   ● $v_{gen}$

Figure 7: t-SNE visualization of feature distributions under different ablation settings on IEMCAP6 1) Unconditional diffusion, 2) DGN only guidance, 3) SCN only guidance, 4) DGN+SCN guidance, colors represent different modalities.

| Available | | $l$ | | $v$ | | $a$ | | $l, v$ | | $l, a$ | | $v, a$ | |
|---|---|---|---|---|---|---|---|---|---|---|---|---|---|
| DGN | SCN | ACC | WAF1 | ACC | WAF1 | ACC | WAF1 | ACC | WAF1 | ACC | WAF1 | ACC | WAF1 |
| ✗ | ✗ | 31.2 | 15.4 | 35.6 | 18.7 | 38.4 | 24.5 | 36.5 | 25.9 | 38.9 | 27.6 | 32.7 | 22.4 |
| ✗ | ✓ | 63.3 | 63.1 | 57.4 | 56.6 | 52.5 | 51.2 | 71.9 | 71.2 | 74.3 | 74.1 | 63.8 | 62.9 |
| ✓ | ✗ | 62.4 | 62.6 | 58.2 | 55.9 | 52.4 | 51.8 | 69.5 | 67.9 | 72.8 | 72.5 | 64.4 | 64.0 |
| ✓ | ✓ | 68.0 | 68.2 | 62.5 | 60.3 | 61.2 | 60.6 | 75.8 | 75.8 | 78.0 | 78.1 | 68.7 | 68.7 |

Table 6: Ablation study results comparing model performance with four ablation settings under various modality incomplete scenarios on the IEMOCAP4 dataset.

## H   Effectiveness of AFS

Figure 8 complements the ablation experiments for AFS presented in Table 3, illustrating the accuracy variation trends of models with and without AFS under all incomplete modality conditions. The figure clearly shows that, across all modality-missing scenarios, models incorporating AFS consistently achieve higher accuracy than those without AFS. The reasons for this can be attributed to the following:

**Differing optimization objectives**: The diffusion model and the classifier pursue distinct optimization goals. During simultaneous training (without AFS), their gradient updates may counteract each other or introduce noise, impeding the model's convergence to an optimal solution. **Impact of**

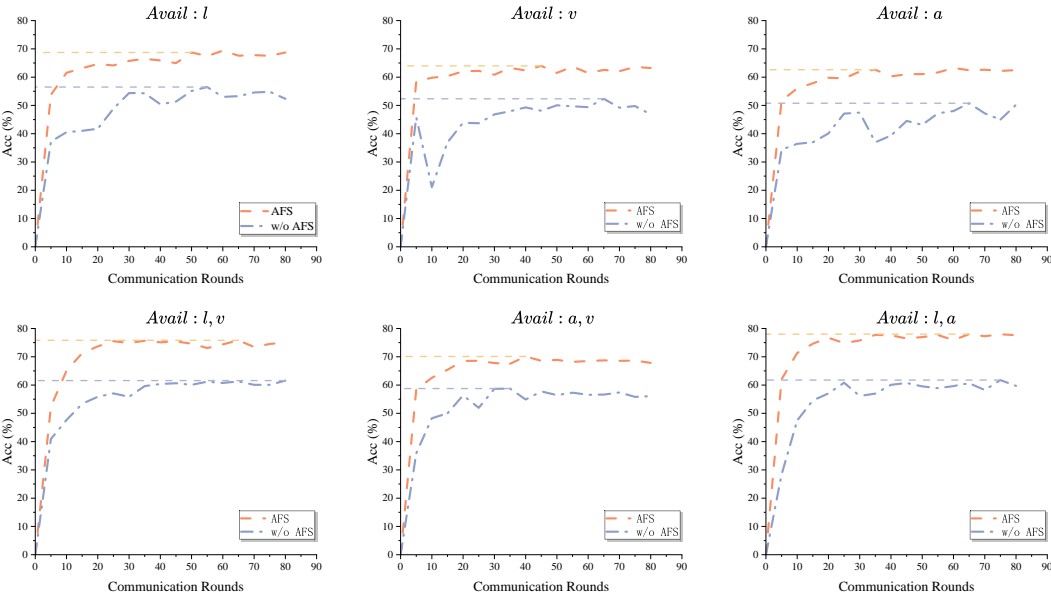

Figure 8: The ACC variation curves of models using AFS compared to models without AFS under all incomplete modality conditions.

**modality recovery quality**: The quality of modality recovery by the diffusion model directly affects the reliability of the input data for the classifier. If the diffusion model is inadequately trained, its outputs may contain residual noise or artifacts, compelling the classifier to learn erroneous feature associations.

AFS addresses these challenges by decoupling the optimization objectives through hierarchical training, which alternately freezes the parameters of the two modules, thereby mitigating gradient conflicts.

## I  Metrics of Different Categories

Figure 9 illustrate the precision and recall, respectively, of FedDISC and other methods for different categories under various incomplete modality conditions on the IEMOCAP4 dataset. The results in **Precision** show that FedDISC (P) achieves superior performance across all subgraphs with the largest enclosed area. Notably, it attains nearly $80\%$ precision on the Anger and Happiness classes under unimodal conditions (e.g., text-only or vision-only) while maintaining competitive accuracy for Sadness and Neutral. In multimodal scenarios, FedDISC (P) further demonstrates strong integration capabilities, significantly outperforming baseline methods. These findings highlight FedDISC's robustness under modality incompleteness. The results in **Recall** show that both FedDISC (I) and FedDISC (P) consistently achieve the highest recall across most emotion categories and all six modality settings, significantly outperforming the three SOTA models even in challenging unimodal or bimodal scenarios with missing modalities. These results highlight FedDISC's superior robustness and generalization capabilities under partial modality conditions, underscoring its effectiveness even when certain modalities are unavailable.

## J  Broader Impacts

This study integrates federated learning with conditional diffusion models to address the dual challenges of modality incompleteness and privacy preservation in multimodal dialogue emotion recognition, enabling technological advancements in healthcare and intelligent customer service. By leveraging distributed collaborative training and modality completion mechanisms, our framework ensures user data privacy while enhancing model adaptability to partially observed modalities. This provides efficient solutions for depression screening and personalized mental health interventions.

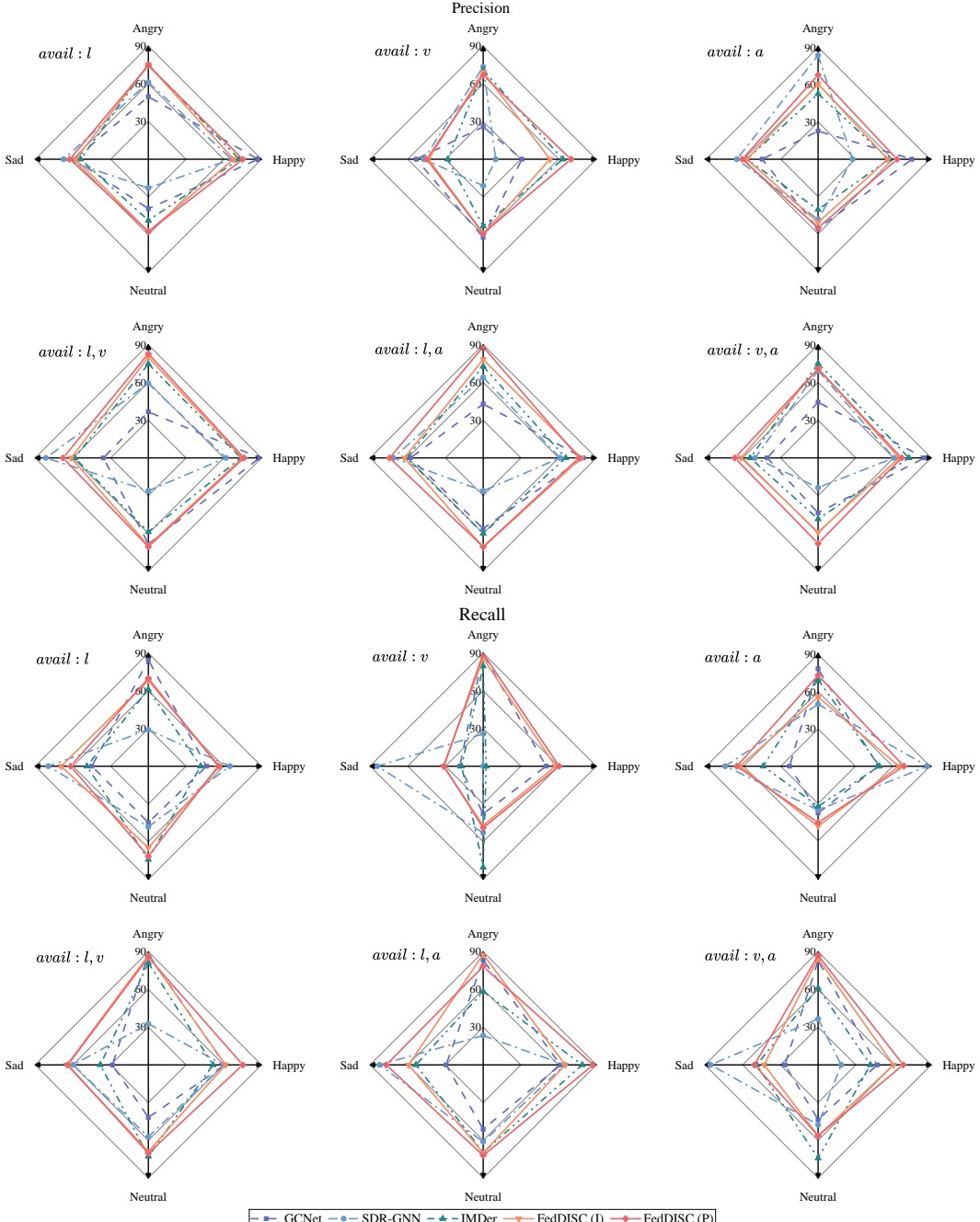

Figure 9: The precision and recall of FedDISC and other methods for different categories under various incomplete modality conditions on the IEMOCAP4 dataset.

Furthermore, the proposed method demonstrates promising applications in cross-institutional medical data sharing, effectively mitigating data silos and advancing AI-driven precision medicine.

Despite its benefits, FedDISC may pose potential risks if deployed without careful consideration. Biases in client-specific training data (e.g., cultural or demographic imbalances in emotion expression) might propagate into the global model, exacerbating fairness issues in cross-client deployments. Moreover, the computational demands of iterative diffusion sampling and federated aggregation could disproportionately exclude resource-constrained participants, reinforcing inequities in collaborative learning ecosystems. Mitigating these risks requires rigorous audits of recovered modality fidelity, fairness-aware aggregation strategies, and energy-efficient implementations to align with ethical AI practices.

