# OpenReview forum: "Federated Dialogue-Semantic Diffusion for Emotion Recognition under Incomplete Modalities"
_NeurIPS.cc/2025/Conference — NeurIPS 2025 poster_

### Official Review · Reviewer_rCdQ · 2025-06-24

**Clarity:** 3
**Significance:** 2
**Originality:** 2
**Rating:** 3
**Confidence:** 4

**Summary:**

1.Using a federated learning framework to address the problem of missing modalities also ensures data privacy.
2. The Federated Dialogue-guided and Semantic-Consistent Diffusion (FedDISC) framework uses a diffusion model to ‌recover missing modality features.

**Questions:**

1. Add some comparative experiments with recent state-of-the-art methods, such as CIF-MMIN[1], MoMKE[2], CM-ARR[3] and Flex-MoE[4] to demonstrate the superiority of your methed.
2.The proposed framework is quite complex; it should include more ablation studies to demonstrate the effectiveness of each module within it.
3. A comparative experiment for the full-modality case should be included in the test set evaluation.

**Ethical Concerns:**

["NO or VERY MINOR ethics concerns only"]

**Final Justification:**

some of the concerns have been addressed. raised the score to 3.

**Limitations:**

yes

**Paper Formatting Concerns:**

The reference format should be more consistent and complete.

**Quality:**

2

**Strengths And Weaknesses:**

Strengths:
1.This paper offers a fresh perspective on emotion recognition under incomplete modalities.
2.The appendix includes extensive supplementary content to help readers better understand the material presented in the main text.
3.Get a good results across multiple datasets.

Weaknesses:
1.Lack a comparison with state-of-the-art methods, e.g., CIF-MMIN[1], MoMKE[2], CM-ARR[3] and Flex-MoE[4].
2.Information regarding the data used for pre-training DGN and SCN is missing.
3.An ablation study comparing the contributions of DGN and SCN is missing.
4.Given that federated learning is the main innovation, the paper does not explore the impact of the number of clients on performance.
5.There is a significant performance disparity between the two diffusion models (DDPM and DDIM), but the paper lacks an in-depth investigation into why this occurs.
6.How are the clients simulated in the experiments? Are they distinct machines, or is it a simulated distributed environment on the two L40S GPUs?

---

> ### Author Rebuttal · Authors · 2025-07-28
>
> **Question1:**
>
> Thank you for your valuable suggestions.
>
> 1. On adding recent SOTA baselines:
>
> We appreciate the reviewer’s suggestion to include comparisons with more recent state-of-the-art methods. In the main text, our comparative experiments already cover the 2021 method (MMIN), as well as recent models from 2023 (GCNet, IMDer, DiCMoR) and 2025 (SDR-GNN). Following your advice—and to maintain a concise table size—we will replaced the 2021 MMIN model with the 2024 CIF-MMIN model in Table 1 and Table 2.
>
> Regarding the four models mentioned by the reviewer:
>
> Unfortunately, as no references were provided, we were unable to identify the corresponding paper for MoMKE.
>
> The CM-ARR model does not have open-source code, making replication infeasible in the short term.
>
> Flex-MoE is designed for Alzheimer's disease multimodal detection and is not applicable to multimodal emotion recognition benchmarks.
>
> Therefore, we conducted experiments with CIF-MMIN under exactly the same settings as our main experiments, and will included these results in the revised manuscript. The comparative results on the IEMOCAP4, IEMOCAP6, CMUMOSI, and CMUMOSEI datasets are summarized below:
> | dataset   | method      | l         | v         | a         | l,v       | l,a       | v,a       | l,v,a      |
> |-----------|-------------|-----------|-----------|-----------|-----------|-----------|-----------|------------|
> | iemocap4  | FedDISC(P)  | 68.0/68.2 | 63.3/61.7 | 62.6/62.1 | 75.8/75.8 | 78.0/78.1 | 68.7/68.0 | 78.6/78.4  |
> | iemocap4  | CIF-MMIN    | 58.0/59.3 | 53.3/51.3 | 56.3/58.4 | 73.2/74.0 | 74.3/75.6 | 65.7/66.9 | 78.3/78.5  |
> | iemocap6  | FedDISC(P)  | 56.8/56.5 | 56.0/55.5 | 60.6/59.9 | 57.4/57.6 | 59.4/59.3 | 66.4/66.3 | 64.3/64.7  |
> | iemocap6  | CIF-MMIN    | 54.7/54.3 | 39.7/35.2 | 56.3/58.4 | 51.3/52.1 | 53.8/50.1 | 56.5/56.9 | 61.3/60.2  |
>
> | dataset   | method      | 0         | 0.1       | 0.2       | 0.3       | 0.4       | 0.5       | 0.6       | 0.7       | average    |
> |-----------|-------------|-----------|-----------|-----------|-----------|-----------|-----------|-----------|-----------|------------|
> | CMUMOSI   | FedDISC(P)  | 86.3/86.3 | 85.9/85.8 | 83.3/83.2 | 81.1/81.0 | 79.7/79.6 | 80.2/80.2 | 77.5/77.5 | 75.7/75.8 | 81.2/81.2  |
> | CMUMOSI   | CIF-MMIN    | 84.0/83.6 | 82.5/84.1 | 82.6/82.7 | 80.5/80.9 | 78.4/78.1 | 77.3/77.2 | 74.1/74.2 | 69.8/71.3 | 78.8/78.9  |
> | CMUMOSEI  | FedDISC(P)  | 86.8/86.4 | 86.5/86.1 | 86.4/86.3 | 85.6/84.7 | 84.5/84.3 | 82.5/83.2 | 81.7/82.5 | 81.5/82.2 | 84.4/84.5  |
> | CMUMOSEI  | CIF-MMIN    | 85.8/86.2 | 85.4/85.5 | 85.0/85.3 | 83.1/83.8 | 82.7/82.5 | 80.4/81.1 | 78.5/79.2 | 77.3/77.4 | 82.3/82.6  |
>
> These results further demonstrate the superiority and robustness of our method compared to strong recent baselines.
>
> 2. On ablation studies:
>
> We sincerely thank the reviewer for this constructive suggestion. However, we beg to differ on the point that additional ablation studies are required. We believe that our paper already provides comprehensive ablation experiments which sufficiently demonstrate the effectiveness of all key modules:
>
> 1\) DGN & SCN:
>
> As shown in Figure 4 of the main text, we compare conditional modality recovery (with DGN and SCN) versus unconditional recovery. This clearly illustrates that DGN and SCN together enable the model to capture more effective semantic information and guide the recovery process. Furthermore, Table 6 and Figure 7 in the appendix provide detailed metric-based and visualization-based ablations, demonstrating that SCN extracts more effective semantic information than a simple speaker graph network.
>
> 2\) DISC-Diffusion:
>
> Figure 3 in the main text and Figure 6 in the appendix visualize the similarity between recovered and original modalities, showing that our DISC-Diffusion module achieves the closest alignment with the real data distribution.
>
> 3\) AFS Module:
>
> Table 3 in the main text quantitatively demonstrates the significant accuracy improvement contributed by the AFS module under various missing-modality scenarios. Moreover, Figure 8 in the appendix shows that AFS not only improves final performance but also accelerates convergence during training.
>
> In summary, we respectfully submit that the ablation studies presented in our manuscript are sufficient to validate the necessity and contribution of each module in our framework.
>
> **Question2:**
>
> Thank you for your valuable suggestion.
>
> We would like to clarify that in Table 2, the results under “missing rate = 0” already correspond to the case where the test set is fully available with all modalities. For further clarity, we will also add the full-modality results to Table 1, as shown below:
>
> | dataset  | available | baseline  | FedDISC(P) | FedDISC(I) | GCNet     | SDR-GNN   | IMDer     | DiCMoR    | CIF-MMIN   |
> |----------|-----------|-----------|------------|------------|-----------|-----------|-----------|-----------|------------|
> | iemocap4 | l,v,a     | 78.6/78.4 | 78.6/78.4  | 78.6/78.4  | 78.4/78.3 | 78.5/78.1 | 78.1/78.3 | 78.2/78.3 | 78.3/78.5  |
> | iemocap6 | l,v,a     | 64.3/64.7 | 64.3/64.7  | 64.3/64.7  | 58.6/59.1 | 61.3/61.2 | 58.7/58.4 | 55.9/56.2 | 61.3/60.2  |
>
>
> The reason we did not originally include full-modality results in Table 1 is that the primary focus of our work is on handling missing modalities, and when all modalities are available, there is no need for modality recovery. What's more, we note that in the recommended baseline in question 1: CIF-MMIN (“Contrastive Learning Based Modality-Invariant Feature Acquisition for Robust Multimodal Emotion Recognition With Missing Modalities”), the authors also do not report full-modality results in their fixed missing-modality experiments. This supports our initial design.
>
> In summary, for completeness, we will add the full-modality case in Table 1 in our revised manuscript. Thank you again for your valuable suggestion.
>
> **Extension:**
>
> We provide responses here to the issues raised by the reviewer in the "Strengths and Weaknesses" . Weakness 1 and Weakness 3 have already been addressed in our previous responses above.
>
> **Weakness2:** Information regarding the data used for pre-training DGN and SCN is missing.
>
> Thank you for pointing out this question. To better simulate real-world scenarios, each client pre-trains its own DGN and SCN modules. The data partitioning for pre-training DGN and SCN is identical to that of the main network.
>
> **Weakness4:** Given that federated learning is the main innovation, the paper does not explore the impact of the number of clients on performance.
>
> Thank you for this question. Regarding the impact of the number of clients, please refer to our response to Question 2 from Reviewer 12cs.
>
> **Weakness5:** There is a significant performance disparity between the two diffusion models (DDPM and DDIM), but the paper lacks an in-depth investigation into why this occurs.
>
> Thank you for your question. The performance difference between the two diffusion models (DDPM and DDIM) is primarily because DDPM enables more accurate feature recovery compared to DDIM. As analyzed in Section 4.2 “Comparison with SOTA Methods” (Visualization experiments), the visualizations demonstrate that features recovered by DDIM deviate more from the original feature distribution, while features recovered by DDPM are much closer to the original distribution. This results in higher downstream accuracy when using DDPM for modality recovery.
>
> **Weakness6:** How are the clients simulated in the experiments? Are they distinct machines, or is it a simulated distributed environment on the two L40S GPUs?
>
> Thank you again for your question. In our federated learning experiments, we follow the classic FedAvg setup: all clients are simulated on the same GPU. Uniquely, we assign the server operations to a separate GPU. For further implementation details, please refer to our upcoming open-sourced code.

---

### Official Review · Reviewer_nTQZ · 2025-06-25

**Clarity:** 3
**Significance:** 2
**Originality:** 3
**Rating:** 4
**Confidence:** 3

**Summary:**

The paper proposes FedDISC, a novel framework for Multimodal Emotion Recognition in Conversations (MERC) that addresses the challenge of missing modalities in real-world scenarios. Unlike conventional methods that rely on complete multimodal data, FedDISC integrates federated learning with diffusion models to recover missing modalities across distributed clients. The DISC-Diffusion module ensures semantic consistency between recovered and available modalities through dialogue-aware modeling and semantic conditioning. An Alternating Frozen Aggregation strategy enables collaborative training of recovery and classification modules. Experiments on IEMOCAP, CMUMOSI, and CMUMOSEI datasets show that FedDISC outperforms existing approaches under various missing-modality conditions.

**Questions:**

1. The authors are encouraged to better clarify the distinctions and advantages of their proposed method compared to previous approaches in the related work section. In particular, a more detailed discussion of existing federated learning methods in multimodal learning is needed to better position the novelty and contributions of FedDISC.
2. Under the fixed missing protocol, where one modality is completely absent from all samples, why the model still performs modality generation instead of simply relying on the available modalities?
3. Why does FedDISC perform worse than SDR-GNN on IEMOCAP6 under the fixed missing setting when the textual modality is present?
4. How do the DGN and SCN modules compare to previous methods that capture context, speaker dependencies, or cross-modal semantics?

**Ethical Concerns:**

["NO or VERY MINOR ethics concerns only"]

**Final Justification:**

The reubttal of authors has addressed Questions 1, 3, and 4. Regarding Question 2, generating missing modalities from existing ones does not introduce new information, and the observed performance gain remains superficial. The authors still lack a deeper, mechanistic explanation for why generation improves results, so I keep the overall scores as it is.

**Limitations:**

The integration of federated learning with diffusion models increases computational complexity and may lead to higher training costs and latency.

**Quality:**

3

**Strengths And Weaknesses:**

Strengths:
FedDISC integrates federated learning with diffusion models to enable collaborative missing-modality recovery while preserving data privacy.
DISC-Diffusion ensures context, speaker, and semantic consistency between recovered and existing modalities using dialogue-aware and semantic conditioning networks.

Weaknesses:
The integration of federated learning with diffusion models increases computational complexity and may lead to higher training costs and latency.

---

> ### Author Rebuttal · Authors · 2025-07-27
>
> **Question1:**
>
> We sincerely thank the reviewer for their insightful feedback and valuable suggestion to enhance the clarity of our related work section. In response, we will revise Section 2.2 (Federated Learning, Pages 2-3) to provide a more comprehensive discussion of existing federated learning (FL) methods in multimodal learning, focusing on their approaches to handling missing modalities, their advantages, and limitations. The revised section will maintain the same word count as the original to ensure consistency with the manuscript’s structure. Below, we outline the planned content for the revised Section 2.2, which will better position the novelty and contributions of our proposed FedDISC framework.
>
> 2.2 Incomplete Multimodal Federated Learning
> Multimodal federated learning(MFL) enables decentralized cross-modal learning while preserving data privacy; however, the inevitable presence of missing modalities in real-world scenarios poses significant challenges to its effectiveness.  Le et al. propose Multimodal Federated Cross Prototype Learning (MFCPL) "Cross-modal prototype based multimodal federated learning under severely missing modality" to addresses the issue of missing modalities through prototype learning, they introduce Cross-Modal Alignment (CMA) to align zero-padded features of missing modalities with existing modality features, reducing noise from zero-padding. However, this method can only deeply mine existing data at the prototype level, failing to fundamentally reconstruct missing information. To tackle this, Yin et. al. "Self-attention fusion and adaptive continual updating for multimodal federated learning with heterogeneous data" introduce Stable Diffusion into MFL to recover missing modalities. On image-text datasets with missing image modalities, clients upload text embeddings, and the server generates and extracts image modality features. However, this method deploys a global generative model on the server, unable to generate client-specific local feature image modalities, limiting its generalization. To address these issues, our FedDISC framework fine-tunes modality-specific feature recovery modules for each client’s local characteristics.
>
> **Question2:**
>
> We sincerely thank the reviewer for their insightful question, which allows us to clarify the necessity of modality generation in FedDISC. Under the fixed missing protocol, where one modality is entirely absent, relying solely on available modalities fails to capture sufficient information for robust emotion recognition, leading to degraded performance. This is evident from baseline methods in Table 1 (Page 7), which show significantly lower accuracy and WAF1 scores when using only available modalities. In contrast, FedDISC achieves substantial improvements by generating missing modalities through client-specific diffusion models. In other words, the recovered modality can provide more comprehensive and enriched information.
>
> **Question3:**
>
> We sincerely thank the reviewer for their meticulous observation. We attribute the performance gap compared to SDR-GNN on IEMOCAP6 under the fixed missing setting with the textual modality present to two key factors.
>
> First, the textual modality in IEMOCAP6 contains significantly richer semantic information than audio and visual modalities, diminishing the impact of modality recovery. As shown in Table 1 (Page 7), baselines achieve higher accuracy when the textual modality is available, as text inherently provides more easily extractable semantics. Consequently, the recovery of audio and visual modalities, which is FedDISC’s primary focus (Section 4, Page 4), becomes less critical for final performance when text is present, reducing our model’s relative advantage.
>
> Second, as analyzed in Section 4.2 (Comparison with SOTA Methods, Page 6), FedDISC’s federated learning paradigm splits data across clients, with each client accessing only one-third of the data available to centralized models like SDR-GNN. This data limitation must be considered when comparing performance, as it inherently constrains FedDISC’s training capacity.
>
>
> **Question4:**
>
> We sincerely thank the reviewer for their insightful question. Experimental results demonstrate that DGN and SCN outperform previous methods in capturing semantic information for emotion recognition. Detailed visualizations in Figure 3 (Page 8) and Appendix Figure 6 (Page 18) illustrate their effectiveness. For instance, Figure 3 shows the recvery capability of SGR-GNN and GCNet, which also use graph networks to model dialogue context. However, their modality recovery performance is significantly inferior to FedDISC, confirming DGN’s superior ability to capture context and speaker dependencies (Section 4, Page 4). Similarly, IMDer attempts to capture cross-modal semantics but achieves poorer modality recovery compared to FedDISC, which leverages cross-attention mechanisms to ensure semantic consistency across modalities (Page 5). These results validate SCN’s effectiveness in capturing cross-modal semantics.

---

> > ### Comment · Reviewer_nTQZ · 2025-08-05
> >
> > Thanks for the detailed response. The clarifications provided have addressed my main concerns.

---

> > > ### Author Response · Authors · 2025-08-05
> > >
> > > Thank you for your positive feedback. We truly appreciate your time and constructive comments, which helped us improve our work.

---

### Official Review · Reviewer_12cs · 2025-07-01

**Clarity:** 2
**Significance:** 2
**Originality:** 2
**Rating:** 4
**Confidence:** 4

**Summary:**

This paper introduces FedDISC, a federated learning method for multimodal emotion recognition in conversations (MERC) under missing modality scenarios. The method integrates generative modality recovery via diffusion models with federated model aggregation. Key components include (1) DISC-Diffusion modules leveraging a Dialogue Graph Network (DGN) and Semantic Conditioning Network (SCN) for context and semantic consistency, and (2) an Alternating Frozen Strategy (AFS) to mitigate optimization conflicts during federated updates. Experiments on IEMOCAP, CMU-MOSI, and CMU-MOSEI demonstrate competitive performance under random and fixed missing modality protocols.

**Questions:**

1. Can you report the inference time and total communication cost for diffusion models per round? Is real-time deployment feasible?
2. How does FedDISC scale with the number of clients? Is there any empirical or theoretical evidence for this?
3.  Were standard data splits used in the evaluation, such as fixed Session 5 for IEMOCAP? If not, comparison fairness is undermined.
4. Can you provide effect sizes or statistical significance tests for the main results?
5. Are there plans to extend FedDISC to asynchronous federated learning or real-world settings with heterogeneous clients?

**Ethical Concerns:**

["NO or VERY MINOR ethics concerns only"]

**Final Justification:**

The paper has several strengths. It tackles an important issue and offers a well-structured solution. Although the technical aspects are based on established methods and do not propose any entirely novel methods, some minor modifications are required. These issues are not sufficient to reject the work. In their rebuttal, the authors address questions regarding speed, scalability, and statistical validation, as well as discuss weaknesses. Their responses have motivated me to change my score to borderline accept. I recommend including the additional details from the rebuttal in the final version of the paper.

**Limitations:**

The authors partially address limitations in Section 5. Acknowledging the sampling inefficiency of diffusion models and the lack of asynchronous training support is a good practice. However, practical constraints such as computational cost, deployment feasibility, and generalization to unseen domains are insufficiently discussed.

**Paper Formatting Concerns:**

Major typos.

**Quality:**

2

**Strengths And Weaknesses:**

Strengths:
1. Well-motivated, addressing real-world problem of multimodal data incompleteness with privacy constraints.
2. Federated generative recovery method is a conceptually interesting direction for MERC.
3. The DISC-Diffusion module design (DGN+SCN) is methodologically reasonable for enforcing multidimensional consistency in recovered modalities.

Weaknesses:
1. Most technical components (e.g., R-GCNs, conditional diffusion, federated averaging) are direct adaptations of known methods. Originality lies primarily in system-level integration, rather than algorithmic innovation.
2. Exact encoder architectures, training configurations are not specified in the main paper.
3. No details on the number of denoizing steps during inference are provided.
4. There are ambiguities in notation.
5. Missing explicit clarification as to whether standard splits (e.g., Session 5 as a test for IEMOCAP) were followed or randomized in the cross-validation process.
6. No ablation for computational cost or real-time feasibility.
7. Claims about generalization and practical deployability are premature. No cross-corpus transfer experiments or asynchronous client tests, despite acknowledging this as a limitation.
8. Typo: "spealer-aware" to "speaker-aware" and "to recovery the feature" to "to recover the feature".

---

> ### Author Rebuttal · Authors · 2025-07-27
>
> **Question1 \& Weakness6：**
>
> Thank you for this valuable question. Based on our experimental results and a comparison to recent literature, we believe real-time deployment of our framework is feasible.
>
> Inference Time:
>
> In our experiments, using DDPM (t=1000), the average inference time is approximately 2.13 minutes per epoch. When using DDIM (t=50), this reduces significantly to 5.7 seconds per epoch.
>
> Communication Cost:
>
> The communication cost per client is 10.57 MB/round when using the conditional DDPM. Compared to the traditional federated learning framework, FedAvg "Communication-efficient learning of deep networks from decentralized data." ( McMahan, et al. PMLR, 2017.), which incurs a communication cost of 22.64 MB/round, our proposed hierarchical federated framework successfully reduces the communication overhead by more than half. Notably, the diffusion model parameters account for the majority of this cost—approximately 85%, and our framework allows for any lighter-weight conditional generative model to be used as the recovery module. For example, when using DDIM, the communication cost further decreases to 7.18 MB per round.
> Moreover, the recent paper “FedDiff: Diffusion Model Driven Federated Learning for Multi-Modal and Multi-Clients” (Li et al., IEEE TCSVT 2024) reports that communication costs of 5–12 MB per round per client are fully deployable in real-world scenarios. This strongly supports the practicality and deployability of our approach.
>
> We sincerely thank the reviewer for raising this insightful question. We will include a detailed discussion of the communication cost in Section K of the appendix (“Communication Cost”) in the revised version.
>
> **Question2:**
>
> For scalability, we first provide a theoretical analysis:
> Our hierarchical federated framework, during the training of the recovery module and the classifier module, can be regarded as two independent FedAvg frameworks. Both strictly follow the synchronous federated learning paradigm, which has been well established in both theory and practice to exhibit linear scalability with respect to the number of clients.
> As pointed out in  "Communication-efficient learning of deep networks from decentralized data." ( McMahan, et al. PMLR, 2017.), the per-client communication cost and aggregation operations do not increase with the total number of clients; both the communication and local storage burdens per client are O(1). Only the server’s total bandwidth increases linearly (O(N), where N is the number of clients), while each client’s overhead remains constant, making horizontal scalability straightforward.
>
> We note that, due to the limitations of the dataset, it is not feasible to fairly investigate the effect of increasing the number of clients on model performance in our experiments. For example, when using **four clients** on the MOSI dataset with a missing rate of 0.1, the result is **84.3/84.8**, which is slightly lower than the results with three clients reported in this paper. However, this reduction can be attributed to the fact that distributing the dataset across four clients reduces the amount of local data available to each client. Additionally, variations in client numbers introduce confounding factors that affect the results. Due to the dataset's limitations, it was challenging to isolate and control these variables effectively. We believe this constraint highlights an important direction for future work, where larger datasets or alternative data partitioning strategies could enable a more robust analysis of client scalability.
>
> **Question3 \& Weakness5:**
>
> We thank the reviewer for their concern regarding the use of standard data splits, and its impact on comparison fairness. To clarify, our experiments on IEMOCAP4 and IEMOCAP6 (Sec. 4.1) adopted the five-fold cross-validation protocol from "GCNet: Graph Completion Network for  Incomplete Multimodal Learning in Conversation" (Zheng et al. IEEE TPAMI, 2023), a widely used benchmark in multimodal emotion recognition. Specifically, we randomly divided the IEMOCAP dataset into five folds, using one fold as the test set and the remaining four for training in each iteration, with the final results averaged over the five runs (Table 1).
>
> **Question4:**
>
> We thank the reviewer for requesting effect sizes and statistical significance tests for our main results. To validate FedDISC(P)’s superiority, we conducted Wilcoxon signed-rank tests with the alternative hypothesis set to “greater” (i.e., FedDISC(P) outperforms baselines) on ACC across modality-missing conditions in Tables 1 and 2, accounting for small sample sizes and non-normal distributions.
>
> Table 1 (IEMOCAP4 and IEMOCAP6): We compared FedDISC(P) with baseline across 6 modality-missing conditions using ACC. Results show W = 21.00, p = 0.0156 < 0.05, indicating significant superiority of FedDISC(P). The effect size, computed as rank-biserial correlation r (r = Z / √N, N=6), is r ≈ 0.77, reflecting a large effect (r > 0.5).
>
> Table 2 (CMU-MOSI and CMU-MOSEI): For 8 modality-missing conditions, Wilcoxon tests yield W = 45.00, p = 0.0020 < 0.05, confirming significant differences. The effect size is r ≈ 0.81 (N=8), also indicating a large effect.
>
> **Question5:**
>
> We thank the reviewer for the question regarding plans to extend FedDISC to asynchronous federated learning (FL) and real-world settings with heterogeneous clients. As noted in Sec. 5 (Conclusion and Limitations), our current framework employs synchronous FL. Recognizing the efficient communication potential of asynchronous FL, we plan to extend FedDISC to this paradigm in future work.
>
> Additionally, to enhance real-world applicability, we intend to extend FedDISC to handle heterogeneous client settings. This includes addressing variations in client data distributions and computational capabilities, using strategies such as personalized FL . Thank you again for your insightful feedback.
>
> **Extension:**
>
> We provide responses here to the issues raised by the reviewer in the "Strengths and Weaknesses" . Weakness 5 and Weakness 6 have already been addressed in our previous responses above.
>
> **Weakness2:**  Exact encoder architectures, training configurations are not specified in the main paper.
>
> Thank you for your question. Our modal-specific encoder consists of a pretrained feature extractor followed by a linear layer. Detailed descriptions of the feature extractors used for each modality can be found in Appendix D (Dataset Segmentation and Preprocessing).
>
> **Weakness3:**  No details on the number of denoizing steps during inference are provided.
>
> We appreciate your careful reading of our manuscript. The number of denoising steps for both diffusion models is specified in Section 4.2, “Comparison with SOTA Methods”: for DDPM, we use 1000 steps, and for DDIM, we use 50 steps.
>
> **Weakness4:**  There are ambiguities in notation.
>
> Thank you for pointing this out. We appreciate your attention to detail. We will thoroughly review the manuscript to clarify any ambiguous notations and ensure all symbols and terms are clearly defined in the revised version.
>
> **Weakness8:**  Typo: "spealer-aware" to "speaker-aware" and "to recovery the feature" to "to recover the feature".
>
> Thank you for catching these typographical errors. We will correct "spealer-aware" to "speaker-aware" and "to recovery the feature" to "to recover the feature" in the revised version. We will also carefully check the manuscript for any other spelling or grammatical issues and make all necessary corrections.

---

> > ### Author Response · Authors · 2025-08-05
> >
> > Dear Reviewer,
> >
> > I hope this message finds you well. As the discussion period is nearing its end with less than three days remaining, I wanted to ensure that we have addressed all of your concerns satisfactorily. If there are any additional points or feedback you would like us to consider, please let us know. Your insights are highly valuable to us, and we are eager to address any remaining issues to further improve our work.
> >
> > If our responses have sufficiently resolved your concerns, we would be grateful if you could kindly raise the score in your final decision.
> >
> > Thank you once again for your time and effort in reviewing our paper.

---

### Official Review · Reviewer_pKQY · 2025-07-03

**Clarity:** 3
**Significance:** 3
**Originality:** 3
**Rating:** 4
**Confidence:** 3

**Summary:**

This paper addresses the problem of missing modalities in multimodal dialogue emotion recognition by proposing the novel FedDISC framework, which pioneers the integration of federated learning into missing-modality recovery. During recovery, it leverages pretrained Dialogue Graph Network (DGN) and Semantic Conditioning Network (SCN) to enforce consistency between recovered and available modalities along the three dimensions of context, speaker identity, and semantics. Moreover, it introduces an Alternating Frozen Strategy to resolve cross-modal optimization conflicts. The manuscript is logically coherent and rigorously organized, and its advantages are validated through comprehensive experiments.

**Questions:**

1:  The framework is complex—how much higher is its computational cost compared to other architectures? If this overhead remains within an acceptable range, I believe the framework’s advantages are indeed significant.

2:  Provide a concise, end-to-end overview of Figures 1 and 2; I am confident this will help readers grasp the workflow more clearly.

3:  In the ablation study, only 120 samples were selected for evaluation. Could this sample size be too small and lack sufficient randomness? Furthermore, when designing the ablation experiments for DGN and SCN, is it possible to validate their individual contributions separately?

4:  After recovering the missing modalities, have the authors considered the potential issue of information redundancy?

**Ethical Concerns:**

["NO or VERY MINOR ethics concerns only"]

**Final Justification:**

Thank you for your response. After reading comments from other reviewers and the responses from authors, I decide to keep my original score.

**Limitations:**

Yes

**Paper Formatting Concerns:**

No paper formatting concerns

**Quality:**

3

**Strengths And Weaknesses:**

### Strengths:
1: The paper is highly innovative, successfully introducing federated learning into missing-modality recovery and offering a new approach for multimodal dialogue emotion recognition under incomplete modalities.

2:  The manuscript is logically rigorous in structure, and demonstrates the method’s strengths through experiments with different missing-modality configurations, varying missing rates, and comprehensive ablation studies.

### Weaknesses:
1:  The framework is complex, and its computational cost relative to other methods remains unclear.

2:  After presenting the methodology in Section 3, a concise overview of the end-to-end workflows shown in Figures 1 and 2 would further aid reader comprehension.

3:  In the ablation study, only 120 samples were selected for evaluation. Could this sample size be too small and lack sufficient randomness? Furthermore, when designing the ablation experiments for DGN and SCN, is it possible to validate their individual contributions separately?

---

> ### Author Rebuttal · Authors · 2025-07-26
>
> **Question1 \& Weakness1：**
>
> Thank you for your thoughtful question regarding computational overhead. We provide a detailed clarification as follows:
>
> Computational Cost:
>
> The primary source of computational overhead in FedDISC lies in the inference time of the diffusion-based recovery module. In our experiments, using DDPM (t=1000), the average inference time is approximately 2.13min/epoch; for DDIM (t=50), this reduces significantly to 5.7s/epoch. As a result, the total training time varies with the diffusion backbone. Importantly, our feature recovery module is not restricted to diffusion models—any conditional generative model with lower inference latency (e.g., conditional autoencoders or GANs) can be flexibly adopted within our framework.
>
> Communication Cost:
>
> The communication cost per client is 10.57 MB/round when using the conditional DDPM. Compared to the traditional federated learning framework, FedAvg "Communication-efficient learning of deep networks from decentralized data." ( McMahan, et al. PMLR, 2017.), which incurs a communication cost of 22.64 MB/round, our proposed hierarchical federated framework successfully reduces the communication overhead by more than half. Notably, the diffusion model parameters account for the majority of this cost—approximately 85%, and our framework allows for any lighter-weight conditional generative model to be used as the recovery module. For example, when using DDIM, the communication cost further decreases to 7.18 MB per round. Moreover, the recent paper “FedDiff: Diffusion Model Driven Federated Learning for Multi-Modal and Multi-Clients” (Li et al.,  IEEE TCSVT 2024) reports that communication costs of 5–12 MB per round per client are fully deployable in real-world scenarios. This strongly supports the practicality and deployability of our approach.
>
> We sincerely thank the reviewer for raising this insightful question. We will include a detailed discussion of the communication cost in Section K of the appendix (“Communication Cost”) in the revised version.
>
> **Question2 \& Weakness2:**
>
> Thank you for highlighting the need for a clearer workflow illustration. Below we provide an end-to-end summary aligned with Figures 1 and 2:
>
> Figure 1: The Dialogue Graph Network (DGN) and Semantic Conditioning Network (SCN) are first pretrained to capture conversational and semantic dependencies from available modalities. DGN builds speaker and context graphs to model dialogue flow and identity, while SCN uses cross- and self-attention to extract semantic information.
>
> Figure 2: The training proceeds in alternating phases:1.Recovery Module Training: Each client uses its available modalities, with DGN and SCN guidance, to train a local diffusion model for missing modalities. These models are aggregated on the server and broadcast back to all clients; 2.Classifier Optimization: With global recovery models fixed, clients reconstruct missing features, forming full-modality representations for emotion classification. Classifier parameters are then aggregated in the server.
>
> **Question3 \& Weakness3:**
>
> We appreciate the reviewer’s careful attention to the ablation design.
>
> Our selection of 120 samples (20 per class) for the t-SNE visualization ablation is inspired by the sampling strategy used in prior work:  “Incomplete Multimodality-Diffused Emotion Recognition” (Wang et al., NeurIPS 2023), where the authors use 70 randomly selected samples (10 per class) for a similar ablation. Compared to this, we chose a slightly larger and class-balanced random subset to enhance representativeness and generalizability.
>
> We appreciate your suggestion to further clarify the roles of DGN and SCN. In fact, we have conducted detailed ablation experiments, and the individual and combined contributions of DGN and SCN are reported in Table 6 and Figure 7 in the supplementary material (Section F). The results demonstrate that 1) Compared to DGN, SCN provides more effective semantic guidance for modality recovery. 2) Combining DGN and SCN offers more comprehensive information, further reducing semantic confusion in the recovered modalities.
>
> **Question4:**
>
> Thank you for this important question regarding information redundancy.
>
> Information redundancy refers to a scenario where the recovered modality features are highly correlated with, or merely duplicates of, the existing modalities, thus failing to provide additional or complementary information for the downstream task.
>
> In our framework, the design of the DGN and SCN helps mitigate this issue. Specifically, these modules capture not only local semantic information within each utterance but also incorporate global dependencies, such as speaker relationships and dialogue context, across the entire conversation. As a result, the recovered modalities are not simple "copies" of the available modalities, but rather enriched features that embed complementary and context-aware cues.
>
> This conclusion is supported by the empirical results in Table 1. If severe information redundancy existed, the results with recovered modalities would be similar to those obtained without recovery (i.e., the Baseline). However, in both the IEMOCAP4 and IEMOCAP6 datasets, our models (FedDISC(P) and FedDISC(I)) achieve substantial improvements over the Baseline.
> Specifically, we performed a one-sided Wilcoxon signed-rank test across the six missing modality cases in IEMOCAP4, and found that the performance gain of FedDISC(P) and FedDISC(I) over Baseline are all statistically significant (W=0, p=0.03125 < 0.05).
> Similar trends hold for IEMOCAP6. This strongly indicates that our recovered modalities provide complementary and non-redundant information, leading to real gains in performance.
>
> We will make this clarification more explicit in the revised appendix (Section F DGN and SCN). Thank you again for raising this valuable point.

---

> > ### Author Response · Authors · 2025-08-05
> >
> > Dear Reviewer,
> >
> > I hope this message finds you well. As the discussion period is nearing its end with less than three days remaining, I wanted to ensure that we have addressed all of your concerns satisfactorily. If there are any additional points or feedback you would like us to consider, please let us know. Your insights are highly valuable to us, and we are eager to address any remaining issues to further improve our work.
> >
> > If our responses have sufficiently resolved your concerns, we would be grateful if you could kindly raise the score in your final decision.
> >
> > Thank you once again for your time and effort in reviewing our paper.

---

> > ### Comment · Reviewer_pKQY · 2025-08-08
> >
> > Thank you for your response. After reading comments from other reviewers and the responses from authors, I decide to keep my original score.

---

> > > ### Author Response · Authors · 2025-08-09
> > >
> > > Thank you for your positive feedback. We truly appreciate your time and constructive comments, which helped us improve our work.

---

### Note · Authors · 2025-08-12

Thanks to the ACs and reviewers for the constructive feedback. Our paper targets multimodal emotion recognition in conversations (MERC) under unpredictable—often extreme—missing modalities. Our contributions are: (i) a **federated, modality-specific diffusion training scheme** in which each client trains diffusion models only on its available modalities; the server aggregates them into **global per-modality models** and redistributes them to clients lacking that modality, enabling **zero-shot cross-client recovery** without any local complete-modality data, thus overcoming single-client incompleteness; (ii) **DISC** via a Dialogue Graph Network (DGN) and a Semantic Conditioning Network (SCN) that capture context, speaker, and semantic dependencies, enforcing **semantic consistency** between recovered and original features; and (iii) an **Alternating Frozen Strategy (AFS)** that hierarchically alternates optimization of the recovery and classifier modules to avoid gradient conflict and stabilize convergence and generalization.

During rebuttal we addressed the main concerns:

**Cost & deployability:** First, our communication cost is within deployable ranges. **In computation** , the diffusion recovery module is the main overhead: DDPM (t=1000) averages ≈2.13 min/epoch, while DDIM (t=50) reduces this to ≈5.7 s/epoch. **On communication** , our  AFS substantially reduces bandwidth. With conditional DDPM the cost is ≈10.57 MB/round, and with DDIM it falls to ≈7.18 MB/round, versus FedAvg’s ~22.64 MB/round. Prior work (FedDiff; Li et al., IEEE TCSVT 2024) reports 5–12 MB/round as fully deployable in practice, supporting our approach’s practicality.

**Statistical validity:** We added Wilcoxon signed-rank tests across folds and missing patterns, reporting significant p-values with large effect sizes, supporting the robustness of the gains.

**Data splits:** We matched standard IEMOCAP protocols (five-fold) and used consistent MOSI/MOSEI label mappings, ensuring comparability and reproducibility.

**Ablations & visualization:** We isolated DGN and SCN, ablated AFS, and provided t-SNE analyses showing recovered features closely align with true distributions rather than being redundant.

We are sincerely grateful to all ACs and reviewers for their time and thoughtful suggestions, and we respectfully ask the ACs to give due consideration to our rebuttal—especially in cases where some reviewers were less actively engaged in the discussion.

---

### Decision · Program_Chairs · 2025-09-17

**Decision:**

Accept (poster)

**Comment:**

This paper proposes FedDISC, a novel framework for multimodal emotion recognition in conversations under incomplete modalities. The main contributions are: (1) the integration of federated learning with diffusion-based generative recovery to handle missing modalities across distributed clients; (2) the DISC-Diffusion module, which ensures semantic and contextual consistency in modality recovery using a dialogue graph network and semantic conditioning network; and (3) an alternating frozen strategy that facilitates stable optimization by decoupling the training of recovery and classification modules. The method is evaluated on standard MERC datasets, demonstrating consistent performance gains under various missing-modality settings.

The paper’s strengths lie in its practical motivation, innovative system-level integration, and thorough empirical validation. It addresses a high-impact, real-world problem—modality incompleteness in privacy-sensitive distributed settings—through a federated generative framework, a direction underexplored in prior work. The integration of DGN and SCN to ensure semantic consistency is well-motivated and effective. The experiments are comprehensive, covering both random and fixed missing-modality protocols, and include strong baseline comparisons, t-SNE visualizations, and statistical significance tests. The authors also demonstrate the communication efficiency and scalability of the approach, responding diligently to reviewer concerns.

The main weaknesses noted were the lack of novelty in individual components, some missing implementation details, and the need for clearer distinction from prior work. One reviewer raised concerns about missing ablations, full-modality baselines, and comparison to recent methods. These were convincingly addressed during the rebuttal, with new experiments (e.g., CIF-MMIN comparisons, full-modality results), detailed clarifications, and acknowledgments of limitations.

In conclusion, although some components build on known methods, the cohesive and well-executed integration, coupled with strong empirical performance and real-world relevance, make this a solid contribution to the multimodal and federated learning communities. Having given some consideration, I recommend acceptance, but this decision might need further discussion with SAC.